Journal of Data-centric Machine Learning Research (2024)          Submitted 12/23; Revised 07/24; Published 08/24

# On minimizing the training set fill distance
# in machine learning regression

**Paolo Climaco**[†]                                                                    CLIMACO@INS.UNI-BONN.DE

**Jochen Garcke** [†,‡]                                                                    GARCKE@INS.UNI-BONN.DE

[†]*Institut für Numerische Simulation, Universität Bonn, Germany*
[‡]*Fraunhofer SCAI, Sankt Augustin, Germany*

**Reviewed on OpenReview:** *https: // openreview. net/ forum? id= 8R7l2uZMVp*

**Editor:** Yue Zhao

## Abstract

For regression tasks one often leverages large datasets for training predictive machine learning models. However, using large datasets may not be feasible due to computational limitations or high data labelling costs. Therefore, suitably selecting small training sets from large pools of unlabelled data points is essential to maximize model performance while maintaining efficiency. In this work, we study Farthest Point Sampling (FPS), a data selection approach that aims to minimize the fill distance of the selected set. We derive an upper bound for the maximum expected prediction error, conditional to the location of the unlabelled data points, that linearly depends on the training set fill distance. For empirical validation, we perform experiments using two regression models on three datasets. We empirically show that selecting a training set by aiming to minimize the fill distance, thereby minimizing our derived bound, significantly reduces the maximum prediction error of various regression models, outperforming alternative sampling approaches by a large margin. Furthermore, we show that selecting training sets with the FPS can also increase model stability for the specific case of Gaussian kernel regression approaches.

**Keywords:** fill distance, farthest point sampling, maximum error, dataset selection, coresets

## 1 Introduction

Machine learning (ML) regression models are widely used in applications, where we are in particular interested in molecular property prediction (Montavon et al., 2013; Hansen et al., 2015). One of the main goals of ML regression is to label, with continuous values, pools of unlabelled data points for which the existing labelling methods, e.g., numerical simulations or laboratory experiments, are too expensive in terms of computation, time, or money. To achieve this, a subset of the unlabelled pool is labelled and used to train a ML model, which is then employed to get fast predictions for the labels of points not considered during training. However, the effectiveness of ML regression models is strongly dependent on the training data used for learning. Therefore, the selection of a suitable training set is crucial for the quality of the predictions of the model. Our focus is on selecting data points that result in a good performance for a variety of regression models. This ansatz ensures that

the labelling effort is not wasted on subsets that may only be useful for specific learning models, classes of models, or prediction tasks.

We distinguish between active and passive dataset selection strategies. Active learning (Settles, 2012) involves learning one or several regression models, predicting uncertainties for unlabelled data, based on which the most uncertain ones are selected for labelling and the cycle starts anew, until (qualitative) stopping criteria are fulfilled. Unfortunately, it typically only benefits a specific model or model class and optimizes the performance of the models for a specific learning task, as it exploits the knowledge of the labels to iteratively update the parameters of the models during the selection process. Passive sampling (Yu and Kim, 2010) is based only on the feature space locations. Consequently, it has the potential to offer advantages when considering multiple learning tasks that pertain to the same data, as it is independent of the label values associated with the analyzed data points. We think passive sampling can be further divided into two subclasses: model-dependent and model-agnostic. Model dependent passive sampling strategies are developed to benefit specific learning models or model classes, such as linear regression (Yu et al., 2006), $k$-nearest neighbors, or naive Bayes (Wei et al., 2015), similar to active learning. Contrarily, model-agnostic strategies have the potential to benefit multiple classes of regression models rather than just one. Farthest point sampling (FPS) (Eldar et al., 1994) is a well-established passive sampling model-agnostic strategy for training set selection already employed in various application fields, such as image classification (Sener and Savarese, 2018) or chemical and material science (Deringer et al., 2021). FPS provides suboptimal solutions to the $k$-center problem (Har-Peled, 2011), which involves selecting a subset of $k$ points from a given set by minimizing the fill distance of the selected set, that is, the maximum over the distances between any point in the remaining set and the selected point nearest to it.

Our study aims to investigate theoretically and empirically the impact of minimizing the training set fill distance by FPS for ML regression. For classification tasks, it was shown that minimizing the fill distance of the training set reduces the average prediction error of Lipschitz-continuous classification models with soft-max output layer and bounded error function (Sener and Savarese, 2018). Unfortunately, these results do not carry over to regression tasks, even for simpler Lipschitz-continuous approaches, such as kernel ridge regression with the Gaussian kernel (KRR) or feed-forward neural networks (FNNs). In particular, we provide examples where reducing the training set fill distance does not significantly lower the average prediction error compared to random selection. The benefits of using FPS in regression have been studied in various works (Yu and Kim, 2010; Wu et al., 2019; Deringer et al., 2021), where it was argued that passive sampling strategies such as FPS are more effective than active learning in terms of data efficiency and prediction accuracy. However, these works lack theoretical motivation, relying only on domain knowledge or heuristics.

In this work, we focus on the maximum prediction error, which can be considered as a measure of the robustness of the predictions of a trained machine learning model. A large maximum prediction error signifies that in some regions of the domain of interest, there is a significant deviation between the predicted and actual target values, that is, there are regions of the feature space where the predictions of the trained model are not reliable. The maximum prediction error is a helpful metric in various applications, such as those related to material science and chemistry, where the average error provides an

incomplete evaluation of the predictions of a model (Sutton et al., 2020; Gould and Dale, 2022). The authors of Vishwakarma et al. (2021), mention the maximum prediction error among those metrics that are "key to comparing the performance of different models and thus for developing guidelines and best practices for the successful application of machine learning in chemistry". In Zaverkin et al. (2022) and Huang et al. (2023), the maximum error is considered to evaluate the prediction quality of machine learning models trained to study and explore the chemical or conformational spaces. Moreover, the authors of Gerrard et al. (2020) use the maximum error to evaluate the prediction quality of Nuclear Magnetic Resonance spectroscopy parameters for 3-dimensional chemical structures.

Consequently, we derive an upper bound for the maximum expected prediction error of Lipschitz continuous regression models that is linearly dependent on the training set fill distance. We show that minimizing the training set fill distance significantly decreases the maximum approximation error of Lipschitz continuous regression models. We compare the FPS approach with other model-agnostic sampling techniques and demonstrate its superiority for low training set budgets in terms of maximum prediction error reduction.

Our analysis offers theoretical and empirical results, which set it apart from previous works. Specifically, we extend the theoretical work of Sener and Savarese (2018) from classification to regression, demonstrating that reducing training set fill distance lowers the maximum prediction error of the regression model. Moreover, contrary to Yu and Kim (2010) and Wu et al. (2019), who studied the advantages of using FPS for regression tasks, our findings are supported by mathematical results providing theoretical motivation for what we show empirically. We emphasize that, according to our knowledge, prior research did not detect the relationship between reducing the fill distance of the training set using FPS and decreasing the maximum prediction error of a regression model, neither theoretically nor empirically. In addition, we provide further theoretical examinations and empirical investigations to show supplementary advantages of selecting training sets with the FPS for kernel regression models, exemplified for a Gaussian kernel. Specifically, our findings indicate that employing FPS for selecting training sets enhances the stability of this particular category of models.

## 2 Related work

Existing work concerning model-agnostic passive sampling is mostly related to coresets approaches. Coresets (Feldman, 2019) aim to identify the most informative training data subset, according to some principle. The simplest coreset method is uniform sampling, which randomly selects subsets from the given pool of data points. Importance sampling approaches, such as the CUR algorithm (Mahoney and Drineas, 2009), assign to samples relevance-based weights. Cluster-besed methods such as $k$-medoids and $k$-medoids++ (Mannor et al., 2011), that are adapted version of $k$-means and $k$-means++ (Murphy, 2022), segment the feature space into clusters and select representative points from each cluster. Greedy algorithms iteratively select the most informative data points based on a predefined criterion. Well-known greedy approaches for subset selection are the submodular function optimization algorithms (Fujishige, 2005; Krause and Golovin, 2014), such as facility location (Frieze, 1974) and entropy function maximization (Sharma et al., 2015). Various coresets strategies have also been designed for specific classes of regression models, such

as $k$-nearest neighbours and naive Bayes (Wei et al., 2015), logistic regression (Guo and Schuurmans, 2007), linear and kernel regression with Gaussian noise (Yu et al., 2006) and support vector machines (Tsang et al., 2005). Such approaches have been designed as active learning strategies and could be developed by exploiting the knowledge of the respective model classes, but do not rely on the models predictions. Assuming knowledge of the learning model may even lead to the development of training set selection strategies that are optimal with respect to some notion of optimality, as in the case of linear regression (John and Draper, 1975). Unfortunately, these selection strategies theoretically guarantee benefits only for certain classes of models. In this work, we are interested in passive sampling strategies that are model-agnostic, thus having the potential to benefit multiple classes of regression models rather than just one.

We investigate the benefits of employing the FPS algorithm (Eldar et al., 1994) for training dataset selection. The farthest point sampling is a greedy algorithm that selects elements by attempting to minimize the fill distance of the selected set, which is the maximal distance between the elements in the set of interest and their closest selected element. The work most similar to our is Sener and Savarese (2018), where the authors show that selecting the training set by fill distance minimization can reduce the average classification error on new points for convolutional neural networks (CNNs) with softmax output layers and bounded error function. However, these benefits do not necessarily extend to regression problems, even with simpler Lipschitz algorithms like KRR and FNN, as we illustrate with our experiments. The advantages of using FPS, thus of selecting training sets with a small fill distance, have also been investigated in the context of ML regression. For instance, in Yu and Kim (2010) the authors argue that for regression problems passive sampling strategies, as FPS, are a better choice than active learning techniques. Moreover, in Wu et al. (2019) and Cersonsky et al. (2021), the authors have proposed variations of FPS, and they argue that these can result in more effective training sets. These variations involve selecting the initial point according to a specific strategy rather than randomly, and exploiting the knowledge of labels, when these are known in advance, to obtain subsets that are representative of the whole set in both feature and label spaces. However, these works only demonstrate the advantages of FPS and its variations empirically and do not provide any theoretical analysis to motivate the benefits of using these techniques for regression.

## 3 Problem definition

We now formally define the problem. We consider a supervised regression problem defined on the feature space $\mathcal{X} \subset \mathbb{R}^d$ and the label space $\mathcal{Y} \subset \mathbb{R}$. We assume the solution of the regression problem to be in a function space $\mathcal{M} := \{f : \mathcal{X} \rightarrow \mathcal{Y}\}$, and that for each set of weights $\boldsymbol{w} \in \mathbb{R}^m$ there exists a function in $\mathcal{M}$ associated with it. $\mathcal{M}$ can be interpreted as the space of functions that we can learn by training a given regression approach through the optimization of its weights $\boldsymbol{w} \in \mathbb{R}^m$. Additionally, we consider an error function $l : \mathcal{X} \times \mathcal{Y} \times \mathcal{M} \rightarrow \mathbb{R}^+$. The error function takes as input the features of a data point, its label, and a trained regression model and outputs a real value that measures the quality of the prediction of the model for the given data point. The smaller the error, the better the prediction.

Furthermore, we consider a dataset $\mathcal{D} := \{(\boldsymbol{x}_q, y_q)\}_{q=1}^k \subset \mathcal{X} \times \mathcal{Y}$, $k \in \mathbb{N}$, consisting of independent realizations of random variables $(\boldsymbol{X}, Y)$ taking values in $\mathcal{Z} := \mathcal{X} \times \mathcal{Y}$ with joint probability measure $p_{\mathcal{Z}}$. We study a scenario in which we have only access to the realizations $\{\boldsymbol{x}_q\}_{q=1}^k$, while the labels $\{y_q\}_{q=1}^k$ are unknown, and the goal is to use ML techniques to predict the labels accurately and fast, recovering from data the relation between the random variables $\boldsymbol{X}$ and $Y$. In supervised ML, we first label a subset $\mathcal{L} := \{(\boldsymbol{x}_{q_j}, y_{q_j})\}_{j=1}^b \subset \mathcal{D}$, $b \ll k$, with $q_j \in \{1, 2, \ldots, k\}$ $\forall j$. We then train a regression model $m_{\mathcal{L}} : \mathcal{X} \to \mathcal{Y}$ using a learning algorithm $A(\cdot) : 2^{\mathcal{D}} \to \mathbb{R}^m$ that maps a labelled subset $\mathcal{L} \subset \mathcal{D}$ into weights $\boldsymbol{w} \in \mathbb{R}^m$ determining the learned function $m_{\mathcal{L}} \in \mathcal{M}$ used to predict the labels of the remaining unlabelled points in $\mathcal{U} := \mathcal{D} - \mathcal{L}$. The symbol $2^{\mathcal{D}}$ represents the set of all possible subsets of $\mathcal{D}$. In the following, we renumber the indices $\{q_j\}_{j=1}^b$ associated with the selected set $\mathcal{L}$, and denote them as $j$, that is, $\mathcal{L} := \{(\boldsymbol{x}_j, y_j)\}_{j=1}^b$. Furthermore, given a set $\mathcal{L} := \{(\boldsymbol{x}_j, y_j)\}_{j=1}^b \subset \mathcal{D}$ we define $\mathcal{L}_{\mathcal{X}} := \{\boldsymbol{x}_j\}_{j=1}^b$ and $\mathcal{L}_{\mathcal{Y}} := \{y_j\}_{j=1}^b$.

In several applications the labelling process is computationally expensive, therefore, given a budget $b \ll k$ of points to label, the goal is to select a subset $\mathcal{L} \subset \mathcal{D}$ with $|\mathcal{L}| = b$ that is most beneficial to the learning process of algorithm $A(\cdot)$. In this work we focus on promoting robustness of the predictions, that is, we want to minimize the maximum expected error of the predictions of the labels obtained with the learned function. Specifically, the problem we want to solve can be expressed as follows:

$$\min_{\substack{\mathcal{L} \subset \mathcal{D}, \\ |\mathcal{L}|=b}} \max_{(\boldsymbol{x}, y) \in \mathcal{U}} \mathbb{E}[l(\boldsymbol{x}, y, m_{\mathcal{L}}) | \boldsymbol{x}], \tag{1}$$

In other words, we aim to select and label a training set $\mathcal{L}$ of cardinality $b$, so that the maximum expected error associated to a trained regression model $m_{\mathcal{L}}$ evaluated on the unlabelled points is minimized. We focus on model-agnostic training set sampling strategies that have the potential to benefit various learning algorithms. In particular, we do not optimize the data selection process to benefit only an a priori chosen class of learning models.

## 4 Fill distance minimization by Farthest Point Sampling

Direct computation of the solution to the optimization problem in (1) is not possible as we do not know the labels for the points. To cope with this issue, we derive an upper bound for the minimization objective in (1) that depends linearly on the fill distance of the training set. Afterwards, we describe FPS, which provides a computationally feasible approach to obtain suboptimal solution for minimizing the fill distance.

### 4.1 Effects of a training set fill distance minimization approach.

First, let us introduce the fill distance, a quantity we can associate with subsets of the pool of data points we wish to label. It can be calculated by considering only the features of the data points.

**Definition 1** *Given* $\mathcal{D}_\mathcal{X} := \{\boldsymbol{x}_q\}_{q=1}^k$ *subset of* $\mathcal{X} \subset \mathbb{R}^d$ *and* $\mathcal{L}_\mathcal{X} = \{\boldsymbol{x}_j\}_{j=1}^b \subset \mathcal{D}_\mathcal{X}$, *the* fill *distance of* $\mathcal{L}_\mathcal{X}$ *in* $\mathcal{D}_\mathcal{X}$ *is defined as*

$$h_{\mathcal{L}_\mathcal{X}, \mathcal{D}_\mathcal{X}} := \max_{\boldsymbol{x} \in \mathcal{D}_\mathcal{X}} \min_{\boldsymbol{x}_j \in \mathcal{L}_\mathcal{X}} \|\boldsymbol{x} - \boldsymbol{x}_j\|_2, \tag{2}$$

*where* $\| \cdot \|_2$ *is the* $L_2$*-norm. Put differently, we have that each point* $\boldsymbol{x} \in \mathcal{D}_\mathcal{X}$ *has a point* $\boldsymbol{x}_j \in \mathcal{L}_\mathcal{X}$ *not farther away than* $h_{\mathcal{L}_\mathcal{X}, \mathcal{D}_\mathcal{X}}$.

Notice that the fill distance depends on the distance metric we consider in the feature space $\mathcal{X}$. In this work, for simplicity, we consider the $L_2$-distance, but the following result can be generalized to other distances.

Next, we formulate two assumptions that we use in the theoretical result. The first assumption concerns the data being analyzed and the relationship between features and labels.

**Assumption 2** *We assume there exists* $\epsilon \geq 0$ *such that for each data point* $(\boldsymbol{x}_q, y_q) \in \mathcal{D}$ *we have that*

$$\mathbb{E}\left[ |Y - \mathbb{E}[Y|\boldsymbol{x}_q]| \, \big| \, \boldsymbol{x}_q \right] := \int_\mathcal{Y} |y - \mathbb{E}[Y|\boldsymbol{x}_q]| \, p(y|\boldsymbol{x}_q) dy \leq \epsilon, \tag{3}$$

*where*

$$p(y|\boldsymbol{x}_q) := \frac{p_\mathcal{Z}(\boldsymbol{x}_q, y)}{p_\boldsymbol{X}(\boldsymbol{x}_q)} \quad and \quad p_\boldsymbol{X}(\boldsymbol{x}_q) := \int_\mathcal{Y} p_\mathcal{Z}(\boldsymbol{x}_q, y) dy. \tag{4}$$

*We refer to '$\epsilon$' as the labels uncertainty. Moreover, we assume that*

$$\left| \mathbb{E}\left[Y|\hat{\boldsymbol{x}}\right] - \mathbb{E}\left[Y|\tilde{\boldsymbol{x}}\right] \right| \leq \lambda_p \|\hat{\boldsymbol{x}} - \tilde{\boldsymbol{x}}\|_2, \tag{5}$$

$\forall \, \hat{\boldsymbol{x}}, \tilde{\boldsymbol{x}} \in \mathcal{X}$, *where* $\lambda_p \in \mathbb{R}^+$.

Formula (3) states that given a realization $\boldsymbol{X} = \boldsymbol{x}_q$, the average absolute difference between the variable Y and its conditional expectation, taken over the distribution of Y given $\boldsymbol{x}_q$, is bounded by a positive scalar $\epsilon$. In simpler words, given a data point location $\boldsymbol{x}_q \in \mathcal{X}$ in the feature space, its associated label value is not fixed. Instead, it tends to be concentrated in a small region of the label space around its conditional expected value, whose size is determined by the positive scalar $\epsilon$. Formula (3) models those scenarios where the underlying true mapping between the feature and label spaces is either stochastic in nature or deterministic with error fluctuations of magnitude parameterized by $\epsilon$. The Lipschitz continuity in (5) is an assumption on the regularity of the map connecting the feature space $\mathcal{X}$ with the label space $\mathcal{Y}$. It tells us that if two data points have close representations in the feature space, then the conditional expectations of the associated labels are also close, that is, elements closer in $\mathcal{X}$ are more likely to be associated labels close in $\mathcal{Y}$.

The second assumption concerns the error function used to evaluate the performance of the model and the prediction quality of the model on the training set. Firstly, to formalize the notion that the prediction error of a trained model on the training set is bounded. Secondly, to limit our analysis to error functions that exhibit a certain degree of regularity, which also reflects the regularity of the regression model.

**Assumption 3** *We assume there exist $\epsilon_{\mathcal{L}} \geq 0$, depending on the labelled set $\mathcal{L} \subset \mathcal{D}$ and the trained regression model $m_{\mathcal{L}}$, such that for each labelled point $(\boldsymbol{x}_j, y_j) \in \mathcal{L}$ we have that*

$$\mathbb{E}[l(\boldsymbol{x}_j, Y, m_{\mathcal{L}})|\boldsymbol{x}_j] \leq \epsilon_{\mathcal{L}}. \tag{6}$$

*We consider $\epsilon_{\mathcal{L}}$ as the maximum expected prediction error of the trained model $m_{\mathcal{L}}$ on the labelled data $\mathcal{L}$. Moreover, we assume that for any $y \in \mathcal{Y}$ and $\mathcal{L} \subset \mathcal{D}$ the error function $l(\cdot, y, m_{\mathcal{L}})$ is $\lambda_{l_{\mathcal{X}}}$-Lipschitz and that for any $x \in \mathcal{X}$ and $\mathcal{L} \subset \mathcal{D}$, $l(\boldsymbol{x}, \cdot, m_{\mathcal{L}})$ is $\lambda_{l_{\mathcal{Y}}}$-Lipschitz and convex.*

With (6) we assume that the expected error on the training set is bounded. Moreover, with the Lipschitz continuity assumptions we limit our study to error functions that show a certain regularity. However, these regularity assumptions on the error function are not too restrictive and are connected with the regularity of the evaluated trained model as we show in Remark 5. For instance, the $\lambda_{l_{\mathcal{Y}}}$-Lipschitz regularity and the convexity in the second argument are verified by all $L_p$-norm error functions, with $1 \leq p < \infty$.

With that, we formulate the main theoretical result of this work, which is a theorem that provides an upper bound for the optimization objective in (1), depending linearly on the fill distance of the selected training set.

**Theorem 4** *Given $\mathcal{D} := \{(\boldsymbol{x}_q, y_q)\}_{q=1}^{k} = \mathcal{U} \sqcup \mathcal{L}$ set of independent realizations of the random variables $(\boldsymbol{X}, Y)$ taking values in $\mathcal{Z} := \mathcal{X} \times \mathcal{Y}$ with joint probability measure $p_{\mathcal{Z}}$, trained model $m_{\mathcal{L}} \in \mathcal{M}$ and error function $l : \mathcal{X} \times \mathcal{Y} \times \mathcal{M} \to \mathbb{R}^+$. If Assumptions 2 and 3 are fulfilled, then we have that*

$$\max_{(\boldsymbol{x}, y) \in \mathcal{U}} \mathbb{E}\left[l(\boldsymbol{x}, y, m_{\mathcal{L}})|\boldsymbol{x}\right] \leq h_{\mathcal{L}_{\mathcal{X}}, \mathcal{D}_{\mathcal{X}}} \left(\lambda_{l_{\mathcal{X}}} + \lambda_{l_{\mathcal{Y}}} \lambda_p\right) + \underbrace{\lambda_{l_{\mathcal{Y}}} \epsilon}_{\substack{\text{labels} \\ \text{uncertainty}}} + \underbrace{\epsilon_{\mathcal{L}},}_{\substack{\text{max error} \\ \text{training set}}} \tag{7}$$

*where $h_{\mathcal{L}_{\mathcal{X}}, \mathcal{D}_{\mathcal{X}}}$ is the fill distance of $\mathcal{L}_{\mathcal{X}}$ in $\mathcal{D}_{\mathcal{X}}$, $\epsilon$ and $\lambda_p$ are the labels uncertainty and Lipschitz constant from assumption 2, respectively, $\lambda_{l_{\mathcal{X}}}$ and $\lambda_{l_{\mathcal{Y}}}$ are the Lipschitz constants of the error function, and $\epsilon_{\mathcal{L}}$ is the maximum expected error of the trained model predictions on the labelled set $\mathcal{L}$.*

**Proof** First we want to find an upper bound for $\mathbb{E}\left[l(\tilde{\boldsymbol{x}}, Y, m_{\mathcal{L}})|\tilde{\boldsymbol{x}}\right]$ for each $\tilde{\boldsymbol{x}} \in \mathcal{U}_{\mathcal{X}}$. Fixed $\tilde{\boldsymbol{x}} \in \mathcal{U}_{\mathcal{X}}$, by the definition of the fill distance we know there exists $\boldsymbol{x}_j \in \mathcal{L}_{\mathcal{X}}$ such that $\|\tilde{\boldsymbol{x}} - \boldsymbol{x}_j\|_2 \leq h_{\mathcal{L}_{\mathcal{X}}, \mathcal{D}_{\mathcal{X}}}$.

$$\begin{aligned}
\mathbb{E}\left[l(\tilde{\boldsymbol{x}}, Y, m_{\mathcal{L}})|\tilde{\boldsymbol{x}}\right] &= \int_{\mathcal{Y}} l(\tilde{\boldsymbol{x}}, y, m_{\mathcal{L}}) p(y|\tilde{\boldsymbol{x}}) dy \\
&\leq \int_{\mathcal{Y}} \left|l(\tilde{\boldsymbol{x}}, y, m_{\mathcal{L}}) - l(\boldsymbol{x}_j, y, m_{\mathcal{L}})\right| p(y|\tilde{\boldsymbol{x}}) dy + \int_{\mathcal{Y}} l(\boldsymbol{x}_j, y, m_{\mathcal{L}}) p(y|\tilde{\boldsymbol{x}}) dy \quad (8) \\
&\leq h_{\mathcal{L}_{\mathcal{X}}, \mathcal{D}_{\mathcal{X}}} \lambda_{l_{\mathcal{X}}} + \int_{\mathcal{Y}} l(\boldsymbol{x}_j, y, m_{\mathcal{L}}) p(y|\tilde{\boldsymbol{x}}) dy
\end{aligned}$$

where $\lambda_{l_{\mathcal{X}}}$ is from Assumption 3. The second inequality in (8) follows from the $\lambda_{l_{\mathcal{X}}}$-Lipschitz continuity of the error function. We can bound the remaining term as follows

$$
\begin{aligned}
\int_{\mathcal{Y}} l(\boldsymbol{x}_j, y, m_{\mathcal{L}}) p(y|\tilde{\boldsymbol{x}}) dy \leq & \int_{\mathcal{Y}} \big|l(\boldsymbol{x}_j, y, m_{\mathcal{L}}) - l(\boldsymbol{x}_j, \mathbb{E}\left[Y|\tilde{\boldsymbol{x}}\right], m_{\mathcal{L}})\big| p(y|\tilde{\boldsymbol{x}}) dy \\
& + \int_{\mathcal{Y}} \big|l(\boldsymbol{x}_j, \mathbb{E}\left[Y|\tilde{\boldsymbol{x}}\right], m_{\mathcal{L}}) - l(\boldsymbol{x}_j, \mathbb{E}\left[Y|\boldsymbol{x}_j\right], m_{\mathcal{L}})\big| p(y|\tilde{\boldsymbol{x}}) dy \\
& + \int_{\mathcal{Y}} l(\boldsymbol{x}_j, \mathbb{E}\left[Y|\boldsymbol{x}_j\right], m_{\mathcal{L}}) p(y|\tilde{\boldsymbol{x}}) dy \\
\leq & \; \lambda_{l_{\mathcal{Y}}} \int_{\mathcal{Y}} \big|y - \mathbb{E}\left[Y|\tilde{\boldsymbol{x}}\right]\big| p(y|\tilde{\boldsymbol{x}}) dy \\
& + \lambda_{l_{\mathcal{Y}}} \int_{\mathcal{Y}} \big|\mathbb{E}\left[Y|\tilde{\boldsymbol{x}}\right] - \mathbb{E}\left[Y|\boldsymbol{x}_j\right]\big| p(y|\tilde{\boldsymbol{x}}) dy \\
& + \int_{\mathcal{Y}} \mathbb{E}[l(\boldsymbol{x}_j, Y, m_{\mathcal{L}})|\boldsymbol{x}_j] p(y|\tilde{\boldsymbol{x}}) dy \\
\leq & \; \lambda_{l_{\mathcal{Y}}}\epsilon + \lambda_{l_{\mathcal{Y}}} \int_{\mathcal{Y}} \left(\lambda_p h_{\mathcal{L}_{\mathcal{X}}, \mathcal{D}_{\mathcal{X}}}\right) p(y|\tilde{\boldsymbol{x}}) dy + \int_{\mathcal{Y}} \epsilon_{\mathcal{L}} \, p(y|\tilde{\boldsymbol{x}}) dy \\
\leq & \; \lambda_{l_{\mathcal{Y}}}\epsilon + \lambda_{l_{\mathcal{Y}}} \lambda_p h_{\mathcal{L}_{\mathcal{X}}, \mathcal{D}_{\mathcal{X}}} + \epsilon_{\mathcal{L}}.
\end{aligned}
\tag{9}
$$

The second inequality follows from the $\lambda_{l_{\mathcal{Y}}}$-Lipschitz continuity of the error function and Jensen's inequality, which is used to obtain the conditional expectation in the integrand of the last term. The third inequality follows from the definition of labels uncertainty, the $\lambda_p$-Lipschitz continuity of the conditional expectation of the random variable $Y$ and the assumption that the expected error on the training set is bounded by $\epsilon_{\mathcal{L}}$. The fourth inequality is obtained by taking out the constants from the integrals in the second and third terms and noticing that, from the definition of $p(y|\tilde{\boldsymbol{x}})$ in (4), we have $\int_{\mathcal{Y}} p(y|\tilde{\boldsymbol{x}}) dy = 1$. Since the above inequality holds for each $\tilde{\boldsymbol{x}} \in \mathcal{U}_{\mathcal{X}}$, we have that

$$
\max_{(\boldsymbol{x},y) \in \mathcal{U}} \mathbb{E}\left[l(\boldsymbol{x}, y, m_{\mathcal{L}})|\boldsymbol{x}\right] \leq h_{\mathcal{L}_{\mathcal{X}}, \mathcal{D}_{\mathcal{X}}} \left(\lambda_{l_{\mathcal{X}}} + \lambda_{l_{\mathcal{Y}}}\lambda_p\right) + \lambda_{l_{\mathcal{Y}}}\epsilon + \epsilon_{\mathcal{L}}.
\tag{10}
$$

■

Formula (7) provides an upper bound for the minimization objective in (1) that is linearly dependent on the fill distance of the training set. Note that our derived bound also depends on the labels uncertainty '$\epsilon$'. In particular, the larger the labels uncertainty, the larger the bound for a fixed training set fill distance. Assuming that the maximum error on the labelled data ($\epsilon_{\mathcal{L}}$) is negligible, the smaller the fill distance, the smaller the bound for the maximum expected approximation error on the unlabelled set, conditional to the unlabelled data locations. Although $\epsilon_{\mathcal{L}}$ is typically considered to be small, its presence in the formula suggests that the maximum expected error on the unlabelled set is also dependent on the maximum error of the predictions on the labelled set used for training, thus, on how well the trained model fits the training data. Additionally, the connection between the bound and the regularity of the map connecting the features and the labels, and the chosen error function are highlighted by the presence of the Lipschitz constants $\lambda_p$, $\lambda_{l_{\mathcal{X}}}$ and $\lambda_{l_{\mathcal{Y}}}$ on the right-hand side of (7).

---

**Algorithm 1** Farthest Point Sampling (FPS)

---

**Input** Dataset $\mathcal{D}_\mathcal{X} = \{\boldsymbol{x}_q\}_{q=1}^k \subset \mathcal{X}$ and data budget $b \in \mathbb{N}$, $b \ll k$.

**Output** Subset $\mathcal{L}_\mathcal{X}^{FPS} \subset D_\mathcal{X}$ with $|\mathcal{L}_\mathcal{X}^{FPS}| = b$.

1: Choose $\hat{\boldsymbol{x}} \in \mathcal{D}_\mathcal{X}$ randomly and set $\mathcal{L}_\mathcal{X}^{FPS} = \hat{\boldsymbol{x}}$.

2: **while** $|\mathcal{L}_\mathcal{X}^{FPS}| < b$ **do**

3: $\quad \bar{\boldsymbol{x}} = \arg \max\limits_{\boldsymbol{x}_q \in \mathcal{D}_\mathcal{X}} \min\limits_{\boldsymbol{x}_j \in \mathcal{L}_\mathcal{X}^{FPS}} \|\boldsymbol{x}_q - \boldsymbol{x}_j\|_2$.

4: $\quad \mathcal{L}_\mathcal{X}^{FPS} \leftarrow \mathcal{L}_\mathcal{X}^{FPS} \cup \bar{\boldsymbol{x}}$.

5: **end while**

---

We remark that if we consider the error function to be the absolute value of the difference between true and predicted labels, Theorem 4 holds for all Lipschitz continuous learning algorithms, such as kernel ridge regression with the Gaussian kernel and feed forward neural networks.

**Remark 5** *If the trained model $m_\mathcal{L} \in \mathcal{M}$ is $\lambda_{l_\mathcal{X}}$-Lipschitz continuous, then also the absolute value error function is $\lambda_{l_\mathcal{X}}$-Lipschitz continuous. To see this, fix $y \in \mathcal{Y}$, $\mathcal{L} \subset \mathcal{D}$ and $\boldsymbol{x}, \tilde{\boldsymbol{x}} \in \mathcal{X}$. Then we have*

$$|l(\boldsymbol{x}, y, m_\mathcal{L}) - l(\tilde{\boldsymbol{x}}, y, m_\mathcal{L})| = \big||m_\mathcal{L}(\boldsymbol{x}) - y| - |m_\mathcal{L}(\tilde{\boldsymbol{x}}) - y|\big| \leq |m_\mathcal{L}(\boldsymbol{x}) - m_\mathcal{L}(\tilde{\boldsymbol{x}})|.$$

*Moreover, the absolute value error function is always $\lambda_{l_\mathcal{Y}}$-Lipschitz with $\lambda_{l_\mathcal{Y}} = 1$. As a matter of fact, fixed $\boldsymbol{x} \in \mathcal{X}$, $m_\mathcal{L} \in \mathcal{M}$ and $y, \tilde{y} \in \mathcal{Y}$ we have*

$$|l(\boldsymbol{x}, y, m_\mathcal{L}) - l(\boldsymbol{x}, \tilde{y}, m_\mathcal{L})| = \big||m_\mathcal{L}(\boldsymbol{x}) - y| - |m_\mathcal{L}(\boldsymbol{x}) - \tilde{y}|\big| \leq |y - \tilde{y}|.$$

### 4.2 Selecting training sets with farthest point sampling

Theorem 4 provides an upper bound for the maximum expected value of the error function on the unlabelled data, conditional to the knowledge of the data features. Our aim is to select a training set by minimizing such a bound. Assuming that the value of the maximum prediction error of the trained regression model on the training set is negligible, we can attempt the minimization of the upper bound in (7) by solving the following minimization problem

$$\min_{\substack{\mathcal{L} \subset \mathcal{D}, \\ |\mathcal{L}| = b}} h_{\mathcal{L}_\mathcal{X}, \mathcal{D}_\mathcal{X}}, \tag{11}$$

where $\mathcal{D} := \{(\boldsymbol{x}_q, y_q)\}_{q=1}^k \subset \mathcal{X} \times \mathcal{Y}$ is the pool of data points we want to label, and $\mathcal{L} := \{(\boldsymbol{x}_j, y_j)\}_{j=1}^b$ is the set of labelled points we use for training. The minimization problem in (11) is equivalent to the $k$-center clustering problem (Har-Peled, 2011). Given a set of points in a metric space, the $k$-center clustering problem consists of selecting $k$ points, or centers, from the given set so that the maximum distance between a point in the set and its closest center is minimized, i.e., the fill distance of the $k$ centers in the set is minimized. Unfortunately, the $k$-center clustering problem is NP-Hard (Hochbaum, 1984). However, using farthest point sampling (FPS), described in Algorithm 1, it is possible to obtain sets with fill distance at most a factor of 2 from the minimal fill distance in polynomial

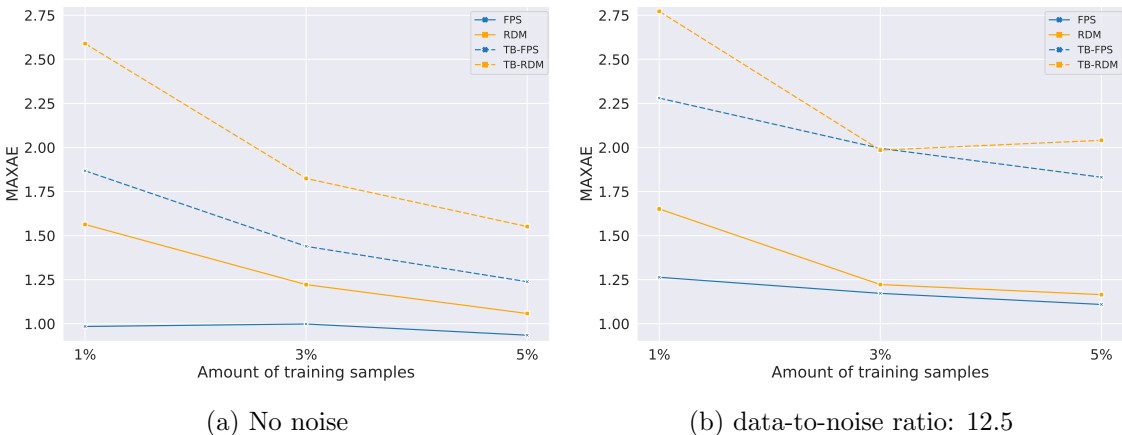

(a) No noise          (b) data-to-noise ratio: 12.5

Figure 1: Results for regression tasks on the illustrative example with a linear regression model trained on sets of various sizes selected randomly and with the FPS. The maximum absolute error (MAXAE) of the trained linear regression model and the theoretical bound (TB) for the expected maximum error of the linear model, computed as in (7), are shown for each training set size. The amount of data used for training is expressed as a percentage of the available data points.

time (Har-Peled, 2011). It is worth to note that reducing the factor of approximation below 2 would require solving an NP-hard problem (Hochbaum and Shmoys, 1985).

The FPS can be implemented using $\mathcal{O}(|\mathcal{D}|)$ space and takes $\mathcal{O}(|\mathcal{D}||\mathcal{L}^{FPS}|)$ time (Har-Peled, 2011). Thus, FPS provides a suboptimal solution, but obtaining a better approximation with theoretical guarantees would not be feasible in polynomial time. To give a qualitative understanding of the data efficiency of FPS, with our implementation of the FPS algorithm, it takes approximately 70 seconds [1] to select 1000 points from the training dataset provided within the selection-for-vision DataPerf challenge (Mazumder et al., 2022), consisting of circa 3.3 millions points in $\mathbb{R}^{256}$.

### 4.3 An illustrative numerical example

To provide a more empirical understanding of our theoretical result and the effects of minimizing the training set fill distance using FPS, we provide an illustrative example where we empirically compute the theoretical bound provided in Theorem 4 and compare it with the computed maximum error achieved by a given regression model.

We analyze a set of data points $\hat{\mathcal{D}} := \{\boldsymbol{x}_i\}_{i=1}^{1000} \subset [-1,1]^2$, the function $f(\boldsymbol{x}_i) = x_{i,1} \times x_{i,2}$, which is the function we aim to predict from data, where $\boldsymbol{x}_i = [x_{i,1}, x_{i,2}]$, and a linear regression model. We consider the absolute error as error function, which is the absolute value of the difference between the actual and predicted function evaluated on a specific data point. We note that $f(\boldsymbol{x})$ is Lipschitz continuous with respect to the absolute error function in $[-1,1]^2$ with Lipschitz constant $\lambda_p = \sqrt{2}$. Moreover, the Lipschitz constant associated with the absolute error of the predictions of a linear regression model are $\lambda_{l_{\mathcal{X}}} = \|\boldsymbol{a}_{\mathcal{L}}\|_2$ and $\lambda_{l_{\mathcal{Y}}} = 1$, where $\boldsymbol{a} \in \mathbb{R}^2$ is the vector of the weights learned by the linear model

---

1. We used a 48-cores CPU with 384 GB RAM.

trained on a set $\mathcal{L} \subset \mathcal{D}$. $\lambda_{l_\mathcal{X}}$ coincides with the Lipschitz constant of the regression model as we know from Remark 5. We also consider a noisy version of the dataset, by adding random noise $\epsilon_i \in \mathbb{R}$ with mean zero and variance 0.1 to each of the data point labels, i.e., $y_i = f(\boldsymbol{x}_i) + \epsilon_i$. To quantify the amount of noise in the data we compute the data to noise ratio $DTNS := \frac{1}{1000} \sum_{i=1}^{1000} y_i^2 \big/ \frac{1}{1000} \sum_{i=1}^{1000} \epsilon_i^2$, which in this illustrative experiment is equal to 12.5. Moreover, we approximate the labels' uncertainty with the maximal noise magnitude $\bar{\epsilon} := \max_{1 \leq i \leq 1000} |\epsilon_i|$.

We train the linear model independently on subsets $\mathcal{L}_j \subset \hat{\mathcal{D}}$, $j = 1, 2, 3$, consisting of 1%, 3% and 5% of the available data points, that is, 10, 30 and 50 points, respectively. Next, we compute the maximal error of the predictions on the training sets, $\{\epsilon_{\mathcal{L}_j}\}_{j=1}^3$, and compute the theoretical bounds (TB) related to each of the selected training sets and respective trained models as follows

$$TB(\mathcal{L}_j, \boldsymbol{a}_{\mathcal{L}_j}) := h_{\mathcal{L}_j, \hat{\mathcal{D}}} \left( \|\boldsymbol{a}_{\mathcal{L}_j}\|_2 + \sqrt{2} \right) + \epsilon_{\mathcal{L}_j} + \bar{\epsilon}.$$

We consider both, training sets selected randomly and with FPS. Fig. 1 illustrate the maximum error of the predictions of the linear model trained on randomly selected training sets (in orange) and sets selected with FPS (in blue), for the noiseless and noisy data scenarios, respectively. Moreover, Fig. 1 illustrate the related $TB$ (dashed lines) for each training set size and selection strategy considered. The figure suggests that the theoretical bound captures the behavior of the true maximum error as expected from the theory, independently of the selection strategy, the training set size, and whether the data is noisy or not. Furthermore, Fig. 1 indicates that selecting the training set by fill distance minimization with the FPS reduces the theoretical bound calculated according to (7) in Theorem 4 and the maximum error of predictions, with respect to the randomly selected training sets. Comparing Fig. 1a and Fig. 1b, we can see that including noise in data increases the maximum prediction error and the theoretical error bound, independently of the selection strategy employed. This can be expected as noise can distort the true underlying relationship between the input features and the target variable, making the regression task more challenging. It is also important to note that including noise does not only affect the bound in terms of the labels uncertainty, but it also has an effect on other quantities such as the maximum error on the training set or the Lipschitz constant of the trained regression model, which is determined by the learned weights. This is because both the maximum error on the training set and the learned weights depend on the label values considered in the regression task.

Additionally, it is worth highlighting that, with FPS, the maximum prediction error is flat, that is, it converges fast to a plateau value. This is particularly evident in Fig. 1a where noise is not included in the data and where it can be clearly seen that such a phenomenon may not be reflected in our proposed bound, indicating that it may be further improved. In the second paragraph of the empirical section 6.5.2, we empirically investigate on three different datasets why there can be a fast decay of the maximum expected error when using the FPS and how this is related to the data points distribution in the feature space. Nonetheless, Fig 1 suggests that our proposed bound provides an effective qualitative estimate of the expected maximum error and that reducing the training set fill distance benefits the robustness of the trained model.

## 5 Kernel ridge regression with the Gaussian kernel (KRR)

Our theoretical analysis suggests that selecting training sets by fill distance minimization with the FPS leads to a reduction of the maximum expected prediction error of Lipschitz continuous models. We note that our analysis relies on the Lipschitz continuity assumption of the loss function, which is related to the regularity of the trained model under consideration, as suggested by Remark 5. A natural question is whether we can highlight additional benefits of using the FPS for selecting the training set by tightening the assumption on the regularity of the regression model through the consideration of specific regression models. In this work, we investigate the additional benefit of selecting the training set using FPS for kernel regression models with the Gaussian kernel, a class of regression approaches successfully employed in various applications such as molecular and material sciences (Deringer et al., 2021), or robotics (Deisenroth et al., 2015). Besides considering the Lipschitz continuity of the (absolute) error function with this regression model, we also study how selecting the training set with the FPS increases the model stability for this specific class of regression approaches.

Kernel ridge regression is a machine learning technique that combines the concepts of kernel methods and ridge regression to perform non-parametric, regularized regression (Deringer et al., 2021). In this work, we use a Gaussian kernel function. Given two data points $\boldsymbol{x}_q, \boldsymbol{x}_r \in \mathcal{X}$, the Gaussian kernel is defined as follows:

$$k(\boldsymbol{x}_q, \boldsymbol{x}_r) := e^{-\gamma \|\boldsymbol{x}_q - \boldsymbol{x}_r\|_2^2}, \tag{12}$$

where $\gamma \in \mathbb{R}^+$ is a kernel hyperparameter to be selected through an optimization process. Provided a training set $\mathcal{L} = \{(\boldsymbol{x}_j, y_j)\}_{j=1}^b$, the weights $\boldsymbol{\alpha} = [\alpha_1, \alpha_2, \ldots, \alpha_b]^T \in \mathbb{R}^b$ of a KRR model are given by the solution of the following minimization problem

$$\boldsymbol{\alpha} = \arg\min_{\bar{\boldsymbol{\alpha}}} \sum_{j=1}^b (m_{\mathcal{L}}(\boldsymbol{x}_j) - y_j)^2 + \lambda \bar{\boldsymbol{\alpha}}^T \boldsymbol{K}_{\mathcal{L}} \, \bar{\boldsymbol{\alpha}}. \tag{13}$$

Here, $\boldsymbol{K}_{\mathcal{L}} \in \mathbb{R}^{b,b}$ is the kernel matrix, i.e., $\boldsymbol{K}_{\mathcal{L}}(q, r) = k(\boldsymbol{x}_q, \boldsymbol{x}_r)$, and the parameter $\lambda \in \mathbb{R}^+$ is the so-called regularization parameter that addresses eventual ill-conditioning problems of the matrix $\boldsymbol{K}_{\mathcal{L}}$. The scalars $\{m_{\mathcal{L}}(\boldsymbol{x}_j)\}_{j=1}^b$ are the labels predicted by the KRR method associated with the training data locations $\{\boldsymbol{x}_j\}_{j=1}^b$. The analytic solution to the minimization problem in (13) is given by

$$\boldsymbol{\alpha} = (\boldsymbol{K}_{\mathcal{L}} + \lambda \boldsymbol{I})^{-1} \boldsymbol{y} \tag{14}$$

where $\boldsymbol{y} = [y_1, y_2, \ldots, y_b]^T$.

Given the location $\boldsymbol{x} \in \mathcal{X}$ of a new data point, its associated predicted label $y(\boldsymbol{x})$ is defined as follows

$$y(\boldsymbol{x}) := m_{\mathcal{L}}(\boldsymbol{x}) = \sum_{j=1}^b \alpha_j k(\boldsymbol{x}, \boldsymbol{x}_j). \tag{15}$$

### 5.1 Kernel ridge regression with data selected by FPS

To address the question of the Lipschitz continuity of the KRR with the Gaussian kernel in view of Theorem 4 and Remark 5, we give the following lemma to express this established fact in our context:

**Lemma 6** *If the error function is the absolute difference between the true and predicted labels, then the regression function provided by the kernel ridge regression algorithm with the Gaussian kernel is Lipschitz continuous.*

**Proof** Consider the training set features $\mathcal{L}_{\mathcal{X}} = \{\boldsymbol{x}_j\}_{j=1}^b$ and set of learned weights $\boldsymbol{\alpha}_{\mathcal{L}} := [\alpha_1, \alpha_2, \ldots, \alpha_b]^T \in \mathbb{R}^b$ obtained by training the KRR on $\mathcal{L}$. Then, given $\boldsymbol{x} \in \mathcal{X}$ the predicted label $y(\boldsymbol{x})$ provided the KRR approximation function can be computed as follows:

$$y(\boldsymbol{x}) = \sum_{j=1}^b \alpha_j k(\boldsymbol{x}, \boldsymbol{x}_j) = \boldsymbol{\alpha}_{\mathcal{L}}^T \boldsymbol{k}_{\boldsymbol{x}}, \tag{16}$$

where $k(\boldsymbol{x}, \boldsymbol{x}_j) := e^{-\gamma \|\boldsymbol{x} - \boldsymbol{x}_j\|_2^2}$, and $\boldsymbol{k}_{\boldsymbol{x}} := [k(\boldsymbol{x}, \boldsymbol{x}_1), k(\boldsymbol{x}, \boldsymbol{x}_2), \ldots, k(\boldsymbol{x}, \boldsymbol{x}_b)]^T \in \mathbb{R}^b$. Next, considering $\tilde{\boldsymbol{x}}, \hat{\boldsymbol{x}} \in \mathcal{X}$, we have

$$\begin{aligned}
|y(\tilde{\boldsymbol{x}}) - y(\hat{\boldsymbol{x}})| &\le |\boldsymbol{\alpha}_{\mathcal{L}}^T \boldsymbol{k}_{\tilde{\boldsymbol{x}}} - \boldsymbol{\alpha}_{\mathcal{L}}^T \boldsymbol{k}_{\hat{\boldsymbol{x}}}| \\
&\le \|\boldsymbol{\alpha}_{\mathcal{L}}\|_2 \|\boldsymbol{k}_{\tilde{\boldsymbol{x}}} - \boldsymbol{k}_{\hat{\boldsymbol{x}}}\|_2 \\
&= \|\boldsymbol{\alpha}_{\mathcal{L}}\|_2 \sqrt{\sum_{j=1}^b \left( e^{-\gamma \|\tilde{\boldsymbol{x}} - \boldsymbol{x}_j\|_2^2} - e^{-\gamma \|\hat{\boldsymbol{x}} - \boldsymbol{x}_j\|_2^2} \right)^2} \\
&\le \|\boldsymbol{\alpha}_{\mathcal{L}}\|_2 \sqrt{b} \lambda_k \|\tilde{\boldsymbol{x}} - \hat{\boldsymbol{x}}\|_2,
\end{aligned}$$

where $\lambda_k$ is the Lipschitz constant of the function $e^{-\gamma r^2}$, $r \in \mathbb{R}^+$. ∎

## 5.2 Increased numerical stability of Gaussian kernel regression with FPS

Numerical stability in a regression approach is a key factor in ensuring the robustness of the learning algorithm with respect to noise and therefore its reliability. A standard criterion for measuring the numerical stability in case of kernel regression is the condition number of the kernel matrix, $\mathbf{K}_{\mathcal{L}} \in \mathbb{R}^{b,b}$. The condition number of a matrix is defined as

$$cond(\mathbf{K}_{\mathcal{L}}) := \|\mathbf{K}_{\mathcal{L}}\|_2 \|\mathbf{K}_{\mathcal{L}}^{-1}\|_2 = \frac{\lambda_{max}(\mathbf{K}_{\mathcal{L}})}{\lambda_{min}(\mathbf{K}_{\mathcal{L}})}, \tag{17}$$

where $\lambda_{max}(\mathbf{K}_{\mathcal{L}})$ and $\lambda_{min}(\mathbf{K}_{\mathcal{L}})$ are the largest and smallest eigenvalues of $\mathbf{K}_{\mathcal{L}}$, respectively. The smaller the condition number, the more numerically stable the algorithm. For high condition numbers, the numerical computations involving the kernel matrix can suffer from amplification of rounding errors and loss of precision that can lead to numerical instability when performing operations like matrix inversion or solving linear systems involving the kernel matrix. Such phenomena may also lead to instability of the predictions as small variations in the input may lead to significant variations in the output.

To increase the model stability we aim to select a training set that leads to a kernel matrix with a small condition number (17). In the following, we recapture results related to the stability of the kernel matrix that have been collected in Wendland (2004) in the context of numerical mathematics. Our goal is to connect these results with the FPS

and show that FPS can be used to increase the stability of KRR models by reducing the condition number of the kernel matrix. For further information and analysis on the stability of kernel matrices, we direct the reader to Wendland (2004), Chapter 12. From the cited literature, we know that the largest eigenvalue of a kernel matrix is mainly dependent on the number of points we consider and not on how we choose them. In particular, the value of the largest eigenvalue can be bounded as follows

$$\lambda_{max}(\mathbf{K}_{\mathcal{L}}) \leq b \max_{q,r=1,\ldots,b} |\mathbf{K}_{\mathcal{L}}(q,r)|. \tag{18}$$

Thus, the maximum eigenvalue is bounded by a quantity that depends linearly on the number of training samples times the maximal entry of the kernel matrix. Since we are considering Gaussian kernels, the maximal entry of the kernel is bounded. Consequently, the value of the maximal eigenvalue grows at most as fast as the number of points we select, independently of how we choose them.

On the contrary, the value of the smallest eigenvalue is strongly dependent on how we choose the training points. To study that, we use the separation distance, a quantity we can associate to subsets of our pool of unlabelled data points.

**Definition 7** *Given set* $\mathcal{L}_{\mathcal{X}} := \{\boldsymbol{x}_j\}_{j=1}^b \in \mathbb{R}^d$, *the* separation distance *of the points in* $\mathcal{L}_{\mathcal{X}}$ *defined as*

$$s_{\mathcal{L}_{\mathcal{X}}} := \frac{1}{2} \min_{\substack{\boldsymbol{x}_q,\boldsymbol{x}_r \in \mathcal{L}_{\mathcal{X}} \\ q \neq r}} \|\boldsymbol{x}_q - \boldsymbol{x}_r\|_2.$$

*In words, the separation distance is half the minimal distance between two points in* $\mathcal{L}_{\mathcal{X}}$. *Given a training set* $\mathcal{L} \subset \mathcal{D}$ *we define* $s_{\mathcal{L}_{\mathcal{X}}}$ *to be its separation distance.*

With the concept of separation distance in mind, we observe that the value of the smallest eigenvalue of the Gaussian kernel matrix can be bounded from below as (Wendland, 2004)

$$\lambda_{min}(\mathbf{K}_{\mathcal{L}}) \geq C_d \left(\sqrt{2\gamma}\right)^{-d} e^{\frac{-40.71d^2}{(s_{\mathcal{L}_{\mathcal{X}}}^2 \gamma)}} s_{\mathcal{L}_{\mathcal{X}}}^{-d}, \tag{19}$$

where $d \in \mathbb{N}$ is the training data dimension, which is fixed, $\gamma \in \mathbb{R}^+$ is the Gaussian kernel hyperparameter, representing the width of the Gaussian, and $s_{\mathcal{L}_{\mathcal{X}}} \in \mathbb{R}^+$ is the training set separation distance. It is important to notice that the lower bound of the smallest eigenvalue decreases exponentially as the separation distance of the selected set decreases. Consequently, given two training sets of the same size, a small difference in their separation distance may lead to a large difference between the smallest eigenvalue of their corresponding kernel matrices, thus also in condition number and model stability. Note that Theorem 12.3 from Wendland (2004) provides a general lower bound for the minimum eigenvalue for all kernels that are, slightly simplified, a positive definite function that possesses a positive Fourier transform, where the bound depends on the separation distance and properties of the Fourier transform. This result holds, besides for the Gaussian kernel, for example also for kernels from the Matérn class under some conditions on the kernel parameters (Rasmussen and Williams, 2006). Given Formula (19), in order to increase the model stability of the

kernel regression approach with Gaussian kernel, we aim to select a training set that solves the following NP optimization problem

$$\max_{\substack{\mathcal{L} \subset \mathcal{D} \\ |\mathcal{L}| = b}} s_{\mathcal{L}_\mathcal{X}}. \tag{20}$$

Interestingly, the FPS provides sets with separation distance at most a factor of 2 from the maximal separation distance (Eldar et al., 1994). Moreover, to obtain an approximation factor better than 2, an NP problem must be solved. Thus, the FPS provides in polynomial time a solution with an optimal approximation factor to both the problems (11) and (20). Consequently, when we consider kernel regression approaches with the Gaussian kernel, selecting the training set with the FPS leads to more robust and stable models.

## 6 Experimental results

The main focus of this section is to investigate the effects of minimizing the training set fill distance on regression tasks from quantum chemistry, where molecular properties are predicted on the QM7, QM8 and QM9 datasets. In particular, we study the performance of FPS in comparison to several sampling baselines while using two machine learning models for prediction, KRR and FNN. Additionally, we empirically investigate the potential benefit of minimizing the training set fill distance for multivariate regression tasks, that is, regression tasks where the label value to predict is multidimensional. More precisely, we focus on the force-field prediction task on molecular trajectories from the rMD17 dataset. The molecular force-field consists of the per-atom forces in a molecule.

Note that our GitHub repository[2] contains all the code necessary to reproduce the results we present. The repository includes code for downloading, reading, and preprocessing the datasets, our implementation of the FPS, regression models, and evaluation procedures. Furthermore, we have included a Jupyter notebook that reproduces the experiments on QM7, with a runtime of only a few minutes.

### 6.1 Datasets

QM7 (Blum and Reymond, 2009; Rupp et al., 2012) is a benchmark dataset in quantum chemistry, consisting of 7165 small organic molecules with up to 23 atoms including 7 heavy atoms: C, N, O and S. It includes information such as the Cartesian coordinates and the atomization energy of the molecules. We use QM7 for a regression task, where the feature vector for a molecule is the Coulomb matrix (Rupp et al., 2012). The Coulomb matrix is defined as

$$\boldsymbol{C}_{i,j} = \begin{cases} \frac{1}{2} z_i^{2.4} & \text{if } i = j \\ \frac{z_i z_j}{\|\boldsymbol{r}_i - \boldsymbol{r}_j\|_2} & \text{if } i \neq j \end{cases} \tag{21}$$

where $z_i$ is the nuclear charge of the $i$-th atom and $\boldsymbol{r}_i$ is its position. In the case of QM7 each molecule is thereby represented as an element in $\mathbb{R}^{529}$, and the label value to predict is the atomization energy, a scalar value describing amount of energy in electronvolt (eV) required to completely separate all the atoms in a molecule into individual gas-phase atoms.

---

2. at `https://github.com/Fraunhofer-SCAI/Fill_Distance_Regression`

QM8 (Ruddigkeit et al., 2012; Ramakrishnan et al., 2015) is a curated collection of 21,786 organic molecules with up to 8 heavy atoms (C, N, O, and F). For each of the molecules it provides the SMILES representation (Weininger, 1988) together with various molecular properties, such as the lowest two singlet transition energies and their oscillator strength. These molecular properties have been computed considering different approaches. In this study we consider those values computed with hybrid exchange correlation functional PBE0. To generate the molecular descriptors we employ Mordred (Moriwaki et al., 2018), a publicly available library that exploits the molecules' topological information encoded in the SMILES strings to provide 1826 physical and chemical features. To work with a more compact representation, we remove 530 features for which the values across the dataset have zero variance. Thus, each molecule in QM8 is represented by a vector in $\mathbb{R}^{1296}$. Furthermore, we normalize the features provided by the Mordred library, to scale them independently in the interval $(0, 1)$. The label value to predict in the regression task is the lowest singlet transition energy (E1), measured in eV, describing the energy difference between the ground state and the lowest excited state in a molecule when both states have singlet spin multiplicity. It is an important property in understanding the electronic behavior of molecules.

QM9 (Ruddigkeit et al., 2012; Ramakrishnan et al., 2014) is a publicly available quantum chemistry dataset containing the properties of 133,885 small organic molecules with up to nine heavy atoms (C, N, O, F). QM9 is frequently used for developing and testing machine learning models for predicting molecular properties and for exploring the chemical space (Faber et al., 2017; Ramakrishnan and von Lilienfeld, 2017; Pronobis et al., 2018). QM9 contains the SMILES representation (Weininger, 1988) of the relaxed molecules, as well as their geometric configurations and 19 physical and chemical properties. In order to ensure the integrity of the dataset, we have excluded all 3054 molecules that did not pass the consistency test proposed by Ramakrishnan et al. (2014). Additionally, we have removed the 612 compounds that could not be interpreted by the RDKit package (Landrum, 2012). Furthermore, in order to ensure the uniqueness of data points, we have excluded 17 molecules that had SMILES representations that were identical to those of other molecules in the dataset. Following this preprocessing procedure, we obtained a smaller version of the QM9 dataset comprising 130,202 molecules. The molecular representation we employ is based on the Mordred (Moriwaki et al., 2018) library, as for the QM8 dataset, with the difference that in this case we do not normalize the features, to show that our results are independent of the normalization. To work with a more compact representation, we remove 519 features for which the values across the dataset have zero variance. Thus, each molecule in QM9 dataset is represented by a vector in $\mathbb{R}^{1307}$. The label value to predict is the HOMO-LUMO energy, measured in eV, describing the difference between the highest occupied (HOMO) and the lowest unoccupied (LUMO) molecular orbital energies. It is a useful quantity for examining the molecules kinetic stability.

The revised MD17 (Christensen and von Lilienfeld, 2020b) (rMD17) is an updated version of the molecular dynamics dataset (MD17) (Unke et al., 2021) commonly used for developing and testing machine learning models for force-field prediction (Schütt et al., 2017; Gasteiger et al., 2022; Liu et al., 2022). The rMD17 consists of temporal trajectories of various small organic molecules of varying sizes and complexity. The dataset provides information on the Cartesian coordinates, atomic charges, and per atom forces, that is,

the force-field, of each molecule at each time step of the molecules' trajectories. The per atom forces are provided in $\frac{\text{kcal}}{\text{mol} \times \text{ångstrong}}$. We use the rMD17 for a multivariate regression task, in which, using the atoms coordinates we aim at predicting the per atom forces of the molecules over the course of their trajectories. The molecular representation we use for the regression task is the one proposed by Chmiela et al. (2017), consisting, for each molecule, of a matrix $\boldsymbol{D} \in \mathbb{R}^{3n_a \times 3n_a}$ defined as follows

$$\boldsymbol{D}_{ij} = \begin{cases} \|\boldsymbol{r}_i - \boldsymbol{r}_j\|_2^{-1} & \text{if } i > j \\ 0 & \text{if } i \leq j \end{cases}$$

where $\boldsymbol{r}_i \in \mathbb{R}^3$ is the geometrical position of the $i$-th atom and $n_a$ is the number of atoms in the analyzed molecule. In this work, we study the trajectories of the Benzene with 9 atoms, Uracil and Malonaldehyde each consisting of 12 atoms. The trajectories of each of the considered molecules consists of 100,000 time steps.

## 6.2 Baseline sampling strategies

We compare the effects of minimizing the training set fill distance through the FPS algorithm with three coresets benchmark sampling strategies. Specifically, we consider random sampling (RDM), the facility location algorithm and $k$-medoids++. Random sampling (RDM) is considered the natural benchmark for all the other coreset sampling strategies (Feldman, 2019), and consists of choosing the points to label and use for training uniformly at random from the available pool of data points. Facility location (Frieze, 1974) is a greedy algorithm that aims at minimizing the sum of the distances between the points in the pool and their closest selected element. $k$-medoids++ (Mannor et al., 2011) is a variant of $k$-means++ (Arthur and Vassilvitskii, 2007), it partitions the data points into $k$ clusters and, for each cluster, selects one data point as the cluster center by minimizing the distance between points labelled to be in a cluster and the point designated as the center of that cluster. Both, facility location and $k$-medoids++, attempt to minimize a sum of pairwise distances. However, the fundamental difference is that facility location is a greedy technique, while $k$-medoids++ is based on a segmentation of the data points into clusters.

## 6.3 Regression models

In this work we use ML regression models that have been utilized in previous works for molecular property prediction tasks. Specifically, we consider kernel ridge regression with the Gaussian kernel (KRR) (Stuke et al., 2019; Deringer et al., 2021) and feed forward neural networks (FNNs) (Pinheiro et al., 2020). KRR and FNN are of interest to us because of their Lipschitz continuity, which, from Remark 5, we know is a required property to validate our theoretical analysis.

We already described KRR in Section 5 and showed the Lipschitz continuity. The hyperparameters $\gamma$ and $\lambda$ are optimized through the following process: first we perform a cross-validation grid search to find the best hyperparameter for each training set size using subsets obtained by random sampling. Next, the average of the best parameter pair for each training set size is used to build the final model. The KRR hyperparameters are varied on a grid of 12 points between $10^{-14}$ and $10^{-2}$. Note that, we do not use an optimal set of hyperparameters for each selection strategy and training set size. This decision is

made because we aim to analyze the qualitative behaviour of a fixed model, where the only variable affecting the quality of the predictions is the selected training set.

Feed-forward neural networks (Goodfellow et al., 2016) (FNNs) are probably the simplest deep neural networks. Given $\boldsymbol{x} \in \mathcal{X}$ the predicted label, $y(\boldsymbol{x})$, provided by a FNN, with $l \in \mathbb{N}$ layers, can be expressed as the output of a composition of functions, that is,

$$y(\boldsymbol{x}) := \phi_l \circ \sigma_l \circ \phi_{l-1} \circ \sigma_{l-1} \circ \cdots \circ \phi_1(\boldsymbol{x}), \tag{22}$$

where the $\phi_i$ are affine linear functions or pooling operations and the $\sigma_i$ are nonlinear activation functions. Following along (Pinheiro et al., 2020), we set $l = 3$, consider only ReLu activation functions and define

$$\phi_i(\boldsymbol{x}) = \boldsymbol{W}_i \boldsymbol{x} + \boldsymbol{b}_i \tag{23}$$

where the weight matrices $\boldsymbol{W}_i$ and the biases $\boldsymbol{b}_i$ are learned by minimizing the mean absolute error between the true and predicted labels of the data points in the training set. The Lipschitz continuity of FNN and other more advanced neural networks has been shown in the literature (Scaman and Virmaux, 2018; Gouk et al., 2020).

For the multivariate regression task we use the gradient-domain machine learning (GDML) method developed in Chmiela et al. (2017), which we follow to introduce the main idea behind GDML, briefly. See Chmiela et al. (2017) for further information on this regression technique. GDML aims to learn the functional relationship

$$\hat{\boldsymbol{f}}_F : \boldsymbol{x}_i \to \boldsymbol{F}_i$$

between the coordinates $\boldsymbol{x}_i \in \mathbb{R}^{3n_a}$ of the atoms in a given molecule and the per-atom forces $\boldsymbol{F}_i \in \mathbb{R}^{3n_a}$. The GDML method relies on a kernel ridge regression technique with Matérn kernel functions to learn such function relationships from data. Given a training dataset $\mathcal{L} = \{\boldsymbol{x}_j\}_{i=1}^b$, the estimation of the function $\hat{\boldsymbol{f}}_F$ on a data point $\boldsymbol{x}$ representing the per atom location in the cartesian space takes the form

$$\hat{\boldsymbol{f}}_F(\boldsymbol{x}) := \sum_{j=1}^b \sum_{i=1}^{3n_a} (\alpha_{j,i}) \frac{\partial}{\partial x_{j,i}} \nabla k(\boldsymbol{x}, \boldsymbol{x}_j),$$

where $\boldsymbol{x}_j = [x_{j,1}, \ldots, x_{j,3n_a}]$, the parameters $\boldsymbol{\alpha} \in \mathbb{R}^{b \times 3n_a}$ are learned by solving a ridge regression type optimization problem, and the function $k : \mathbb{R}^{3n_a} \times \mathbb{R}^{3n_a} \to \mathbb{R}$ is the employed Matérn kernel function. Given that GDML is based on a differentiable Matérn kernels we expect its predictions to exhibit some regularity. However, analyzing the Lipschitz continuity of this regression model is beyond the scope of this work.

## 6.4 Evaluation metrics

This section introduces and defines the metrics we consider to evaluate the performances of the studied regression models. We consider evaluation metrics for univariate and multivariate tasks, that is, for regression tasks with scalar and vector-valued labels, respectively.

Univariate regression

We consider two metrics to evaluate the performance of the ML methods used for the regression tasks with scalar label values: Maximum Absolute Error (MAXAE) and Mean Absolute Error (MAE). The MAXAE is the maximum absolute difference between the true target values $\{y_i\}_{i=1}^n$ and the predicted values $\{\tilde{y}_i\}_{i=1}^n$, that is,

$$\text{MAXAE} := \max_{1 \leq i \leq n} |y_i - \tilde{y}_i|, \tag{24}$$

where $n$ is the number of unlabelled data points in the analyzed data pool. The MAE is calculated by averaging the absolute differences between the predicted values and the true target values, that is,

$$\text{MAE} := \frac{1}{n} \sum_{i=1}^n |y_i - \tilde{y}_i|. \tag{25}$$

Furthermore, to evaluate the stability of the KRR approach we also consider the condition number of the kernel obtained from the training data defined as in (17).

Multivariate regression

We consider three metrics to evaluate the performance of the ML method used for the force-field regression tasks with vector-valued label values: the atom-wise maximum error over the predicted forces ($\text{MAXAE}_F$), the molecule-wise maximum MAE ($\text{MAXMAE}_F$) and the mean absolute error ($\text{MAE}_F$). The $\text{MAXAE}_F$ is the maximum absolute difference between the entries of the true target values $\{\boldsymbol{F}_i\}_{i=1}^n \subset \mathbb{R}^{3n_a}$, describing the per-atom forces of the analyzed molecule with $n_a$ atoms, and those of the predicted values $\{\tilde{\boldsymbol{F}}_i\}_{i=1}^n \subset \mathbb{R}^{3n_a}$, that is,

$$\text{MAXAE}_F := \max_{1 \leq i \leq n} \max_{1 \leq j \leq 3n_a} |F_{i,j} - \tilde{F}_{i,j}|, \tag{26}$$

where $\boldsymbol{F}_i = [F_{i,1}, F_{i,2}, \ldots, F_{i,3n_a}]$ and the $3n_a$ is due to the fact that for each of the $n_a$ atoms in the molecule we consider the forces along the three Cartesian axes. The $\text{MAXMAE}_F$ is defined as follows:

$$\text{MAXMAE}_F := \max_{1 \leq i \leq n} \left( \frac{1}{3n_a} \sum_{j=1}^{3n_a} |F_{i,j} - \tilde{F}_{i,j}| \right). \tag{27}$$

Both, the $\text{MAXAE}_F$ and the $\text{MAXMAE}_F$, are quantities we introduce to evaluate the robustness of the predictions of a given regression model for the force-field prediction task. The $\text{MAXAE}_F$ provides an atom-wise information on the worst case prediction error while the $\text{MAXMAE}_F$ focuses on the molecule-wise worst case error. To evaluate the average performance of a multivariate regression model we consider the $\text{MAE}_F$, that is, the average absolute differences between the predicted values and the true target values:

$$\text{MAE}_F := \frac{1}{3nn_a} \sum_{i=1}^n \sum_{j=1}^{3n_a} |F_{i,j} - \tilde{F}_{i,j}|. \tag{28}$$

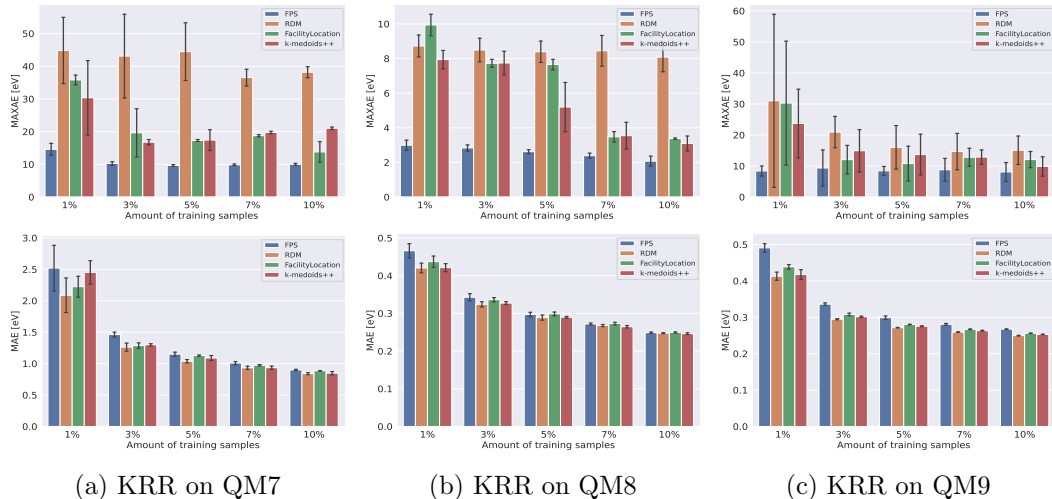

(a) KRR on QM7        (b) KRR on QM8        (c) KRR on QM9

Figure 2: Results for regression tasks on QM7, QM8 and QM9 using KRR trained on sets of various sizes, expressed as a percentage of the available data points, and selected with different sampling strategies. MAXAE (top row) and MAE (bottom row) are shown for each training set size and sampling approach.

## 6.5 Numerical Results

The experiments we perform involve testing the predictive accuracy of each trained model on all data points not used for training, in terms of the MAXAE and MAE for the molecular property prediction task and of the $MAXAE_F$, $MAXMAE_F$ and $MAE_F$ for the force-field prediction task. For each sampling strategy, we construct multiple training sets consisting of different amounts of samples. For each sampling strategy and training set size, the training set selection process is independently run five times. In the case of RDM, points are independently and uniformly selected at each run, while for the other sampling techniques, the initial point to initialize is randomly selected at each run. Therefore, for each selection strategy and training set size, each analyzed model is independently trained and tested five times. The reported test results are the average of the five runs. We also plot error bands, which, unless otherwise specified, represent the standard deviation of the results. We remark that the final goal of our experiments is to empirically show the benefits of using FPS compared to other model-agnostic state-of-the-art sampling approaches. We do not make any claims on the general prediction quality of the employed models on any of the studied datasets.

### 6.5.1 Molecular property prediction on QM7, QM8 and QM9 datasets

Fig. 2 and Fig. 3 show the results for the regression tasks on the QM7, QM8 and QM9 datasets using KRR and FNN, respectively. The graphs on the top rows of Fig. 2 and Fig. 3 illustrate the maximum error of the predictions on the unlabelled points. The results suggest that, independently of the dataset and the regression model employed for the regression task, selecting the training set by fill distance minimization using FPS, we can perform better than the other baselines in terms of the maximum error of the predictions.

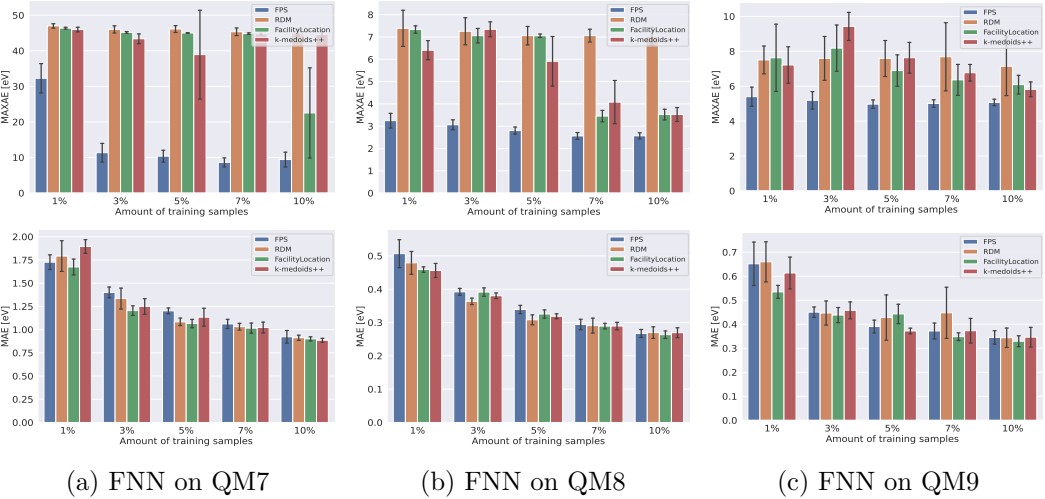

(a) FNN on QM7          (b) FNN on QM8          (c) FNN on QM9

Figure 3: Results regression tasks on QM7, QM8 and QM9 using FNN trained on sets of various sizes, expressed as a percentage of the available data points, and selected with different sampling strategies. MAXAE (top row) and MAE (bottom row) of the predictions are shown for each training set size and sampling approach.

The graphs on the bottom row of Fig. 2 and Fig. 3 show the MAE of the predictions on the QM7, QM8 and QM9 datasets for KRR and FNN, respectively. These graphs indicate that selecting training sets with FPS doesn't drastically reduce the MAE of the predictions on the unlabelled points with respect to the baselines, independently of the dataset and regression model. On the contrary, we observe examples where FPS performs worse than one of the baselines, e.g., with the FNN on QM7, QM8 and QM9 when trained with 5% of the available data points. These experiments suggest that, contrary to what has been shown for classification (Sener and Savarese, 2018), selecting training sets by fill distance minimization does not provide any significant advantage compared to the baselines in terms of the average error. This marks a fundamental difference between regression and classification tasks regarding the benefits of reducing the training set fill distance. The graphs on the top row of Fig. 4 show the condition number of the regularized kernel matrices generated during training of the KRR approach and used to calculate the regression parameters as shown in (14). For QM9, the condition number appears not to be affected by the training dataset choice, while for QM7 and QM8, choosing training sets with FPS reduces the condition number of the regularized kernel, particularly in the low data regime, leading to improved stability of the learned model as discussed in Section 5.2. We remark that the graphs in the top row of Fig. 4 depict the condition number of the regularized kernel matrix $\boldsymbol{K}_{\mathcal{L}}+\lambda\boldsymbol{I}$, where $\boldsymbol{K}_{\mathcal{L}}$ is the Gaussian kernel matrix built from the training data and $\lambda\boldsymbol{I}$ is the regularization term, introduced in (13), used to address ill-conditioning problems. The hyperparameter $\lambda$ has been chosen following a procedure based on cross-validation on randomly selected subsets of the available data pool, as explained in Section 6.3. The values of $\lambda$ we employed are $1.9 \cdot 10^{-4}$, $2.2 \cdot 10^{-3}$ and $1.5 \cdot 10^{-11}$ for QM7, QM8 and QM9, respectively. The bottom row of Fig. 4 illustrates the condition numbers of the non-regularized kernels. From the figure it can be clearly seen that, if no regularization is applied, for QM7 and QM8 the

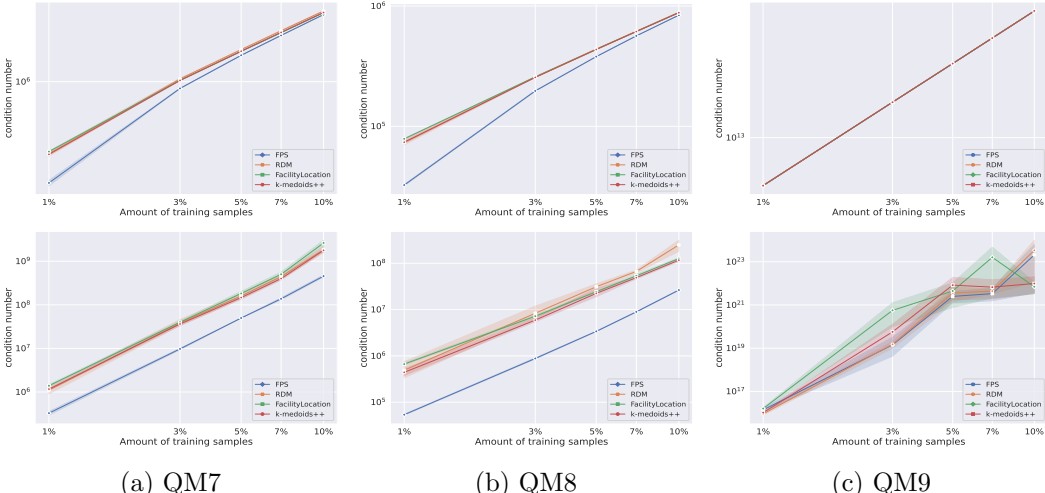

(a) QM7          (b) QM8          (c) QM9

Figure 4: Condition number of the regularized (top row) and non-regularized (bottom row) Gaussian kernels are shown for each dataset, training set size and sampling approach. The graphs are on log-log scale and the error bands represent the confidence interval over five independent runs of the experiments.

difference between the condition numbers of the matrices obtained with the FPS and those obtained using the benchmark strategies is close to an order of magnitude, as expected from (19). As for QM9, we still see a lower condition number when using the FPS in the low data limit, until 7% of the data is employed for training. Notice that for QM9, the magnitude of the condition number is significantly higher than for the other datasets due to the larger size of the kernel matrix. It is also important to mention that, in our experiments differences in the conditions numbers are mainly due to differences in the minimum eigenvalues, in line with the theory reported in Section 5.2.

### 6.5.2 Empirical analysis and discussion

This section further examines the empirical results presented in Section 6.5.1. Specifically, we identify the overall trends of the MAXAE and relate them to our theoretical study, focusing on their connection with the concept of fill distance of the training set. We emphasize what we think should be the practical application of our theoretical result. Next, we analyze the benefit of employing the FPS from a more empirical perspective, focusing on understanding how the FPS selection process works, that is, what points are prioritized during the selection process, how they are distributed, and what consequence this has on the learning process of a given regression model. Next, based on the observations of our empirical study, we discuss the limitations of the FPS.

Interestingly, with FPS, the MAXAE converges fast to a plateau value for all datasets and regression models (Figs. 2 and 3). Differently, with the baseline approaches, the MAXAE has much larger values in the low data regime and tends to decrease gradually as the size of the training sets increases. It is important to notice that these trends of the MAXAE of the predictions are directly correlated with the fill distances of the respective labelled sets used for training, illustrated in Fig. 5a. From Fig. 5a it can be clearly seen

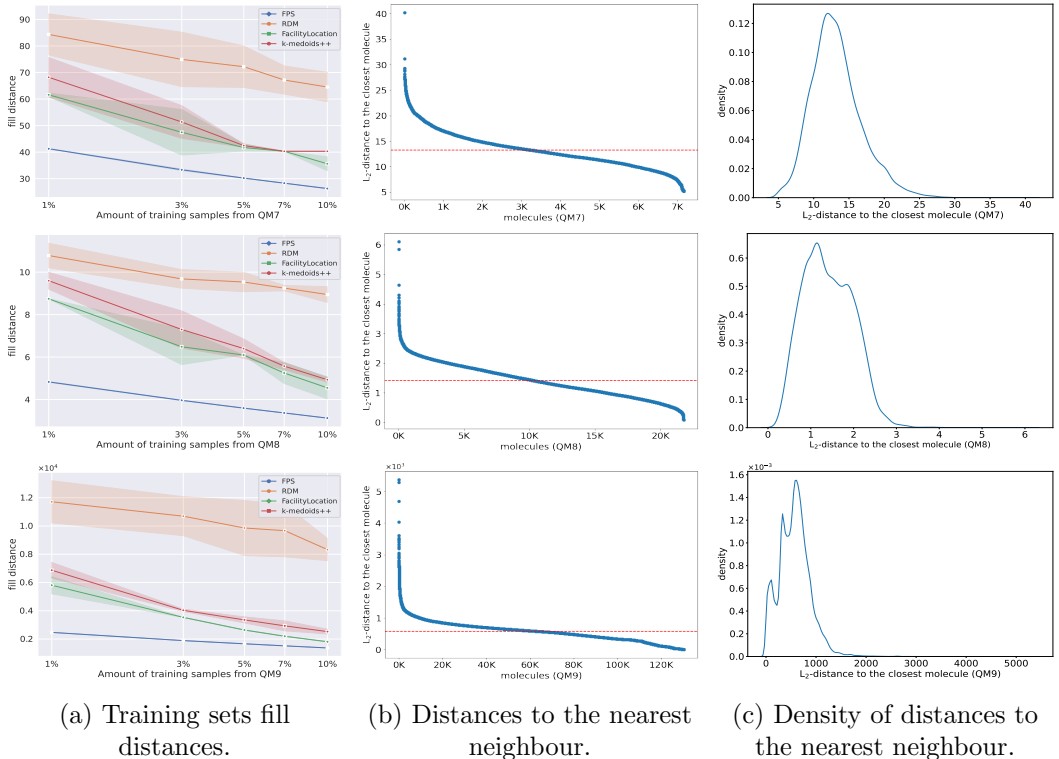

(a) Training sets fill distances.

(b) Distances to the nearest neighbour.

(c) Density of distances to the nearest neighbour.

Figure 5: (a) Fill distances of the selected training sets. (b) Euclidean distances to the nearest neighbour and (c) density of such distances for molecules in QM7 (top row), QM8 (middle row) and QM9 (bottom row). In (b) the red lines are the average distances between the molecules in the datasets and their nearest neighbour and the molecules are sequentially numbered such that the distances decrease in magnitude as the associated molecule numbers increase.

that independently of the dataset considered, with FPS, the fill distances are consistently lower even for small data budgets, while with the benchmarks, the fill distances are much larger in the low data regime and gradually decrease as the size of the training set increases. These observations indicate that the training set fill distance is directly correlated with the maximum error of the predictions on the unlabelled set. Consequently, by minimizing the training set fill distance, we can drastically reduce the MAXAE of the predictions. Nevertheless, our theoretical analysis shows that the training set fill distance is only linked to the maximum expected value of the error function computed on the unlabelled points. Moreover, this bound also depends on other quantities we may not know or that we cannot compute a priori. Namely, the labels uncertainty and the maximum prediction error on the training set, quantifying how well the trained regression model fits the training data. Thus, we believe that the training set fill distance should not be considered as the only parameter to obtain an a priori quantitative evaluation of the MAXAE of the predictions, but as a qualitative indicator of the model robustness that, if minimized, leads to a substantial reduction of the MAXAE.

As we mentioned, the bound provided in Theorem 4 also depends on the label's uncertainty and the maximum error on the training set. Thus, it is worth to provide additional insight on these two quantities. Unfortunately, the labels uncertainty is an intrinsic property of the dataset that we cannot compute or estimate unless we know the true solution of the regression problem or we have an estimate of the error performed by the numerical procedure used to create the dataset. Nonetheless, we can perform a-posteriori computation of the maximum error on the training sets and evaluate its behaviors with respect to the different selection strategies and the regression models we consider. Thus, in what follows we analyze the maximum error on the training sets selected for the experiments in Section 6.5.1 related to the best and worst performing selection strategies, that is, we consider FPS and random sampling, respectively.

Fig. 6 shows the maximum absolute prediction error on the training sets for QM7, QM8 and QM9 using KRR (top row) and FNN (bottom row). The training sets we consider are the same we used for the experiments in Figs. 2 and 3. It can be clearly seen that the maximum error tends to increase as the size of the training set increases, which is an expected result since the more data samples we have, the more difficult it becomes for the model to accurately fit all the data points. Moreover, it is worth noticing that the behavior of the maximum error on the training set for the KRR is consistent across different datasets and training set sizes, independently of the selection strategy we consider. In particular Fig. 6 suggests that, for the KRR, maximum errors on the training sets selected with the FPS and randomly are comparable. This observation also holds for the FNN on QM7. This fact indicates that differences in the maximum error on the relative test sets are mainly due to the other quantities appearing in the bound provided in Theorem 4, such as the training set fill distance. However, when comparing the FPS and random sampling (RDM) on QM8 and QM9 with FNN, we notice more pronounced differences. Notably, on the training sets selected with RDM the maximum error tends to be smaller than on those selected with FPS. This may be motivated by the fact that FPS tends to select points that are farther apart and potentially isolated, as we later see, making it more difficult to reach local minima in the nonlinear optimization problem we are required to solve for the optimization of the weights of the FNN. It is also interesting to compare Fig. 6, which illustrates the error on the training sets, with Figs. 2 and 3, which illustrates the error on the relative test sets. From this comparison, we notice that for KRR, the maximum error on the training set is negligible or much smaller compared to the maximum error on the test set, particularly for QM8 and QM9. This is not the case for the FNN regression model. Specifically, with FNN on QM8 and QM9, the maximum error on the training sets selected with FPS is larger than on the test set. We relate this to the presence of isolated points in the training set that are difficult to learn when solving a nonlinear optimization problem.

As a matter of fact, we think that the effectiveness of FPS is also due to its ability to sample, even for small training sets sizes, those points that are at the tails of the data distribution and that are convenient to label, as the predictive accuracy of the learning methods on those points would be limited due to the lack of data information in the portions of the feature space where data points are more sparsely distributed. To see this empirically, let us first consider Fig. 5b and Fig. 5c, showing for each molecule the Euclidean distance to the respective closest molecule and the density of such distances, respectively, for the QM7, QM8 and QM9 datasets. Fig. 5b shows that, in all the analyzed datasets, there are

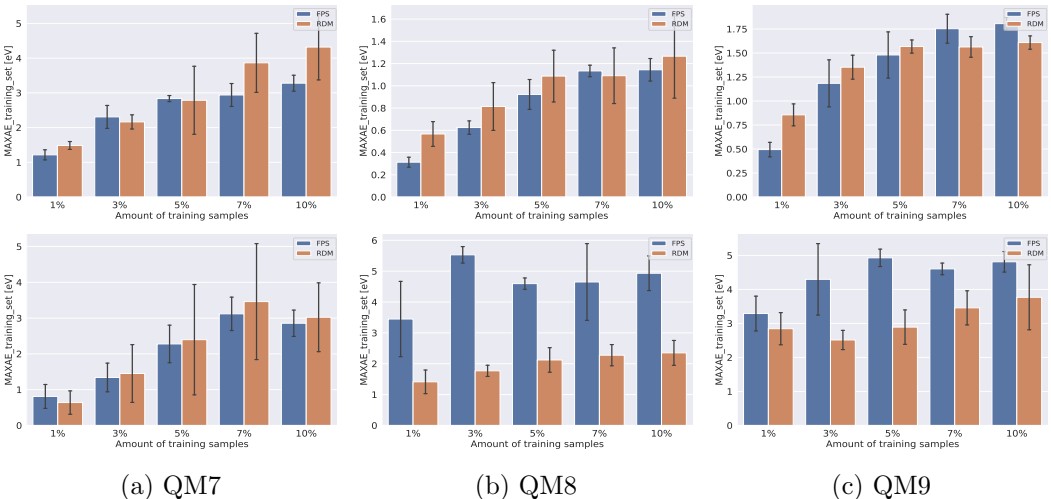

(a) QM7           (b) QM8           (c) QM9

Figure 6: MAXAE on training set for QM7, QM8 and QM9 using KRR (top row) and FNN (bottowm row) trained on sets of various sizes, expressed as a percentage of the available data points, and selected randomly (RDM) or with FPS.

"isolated" molecules for which the Euclidean distance to the nearest molecule is more than twice the average distance between the molecules in the dataset and their nearest neighbor, represented by the red line in the graphs. Fig. 5c, representing the density distribution of the distances of the molecules to their closest data point, tells us that the "isolated" molecules are only a very small portion of the dataset and, therefore, represent the tail of the data distribution. We now see that FPS, contrary to the other baselines, can effectively sample the isolated molecules even for a low training data budget. Fig. 7 highlights the Euclidean distances to the closest neighbour for molecules selected with FPS, and the other baseline strategies, from all the analyzed datasets. The size of the selected sets is 1% of the available data points. Specifically, we are analyzing the same elements selected in the lowest training data budget we considered for the regression tasks in Figs. 2 and 3. Fig. 7 clearly illustrates that, independently of the dataset, FPS selects points across the whole density spectrum. On the contrary, the baseline methods mainly sample points that have a closer nearest neighbour and that are nearer to the center of the data distribution (Fig. 5c).

The observation that selecting isolated molecules is beneficial in terms of the MAXAE reduction is also in line with Theorem 4. We know that a sampling strategy that aims to reduce the maximum error of the predictions should minimize the fill distance of the training set. Thus, it should include the isolated molecules in the training set, as their distance to the nearest neighbour is much larger than the average.

Our empirical analysis indicates that using FPS can be advantageous in the low training data budget, as it allows including early in the sampling process the "isolated" molecules. But, once the data points at the tails of the data distribution have been included, we believe that there may be more convenient sampling strategies than FPS to select points at the center of the distribution, where more information is available. To empirically support the hypothesis that the FPS is mostly beneficial in the low data limit, Fig. 8 illustrates the MAXAE of the predictions on QM7, QM8 and QM9, for the KRR and FNN trained on

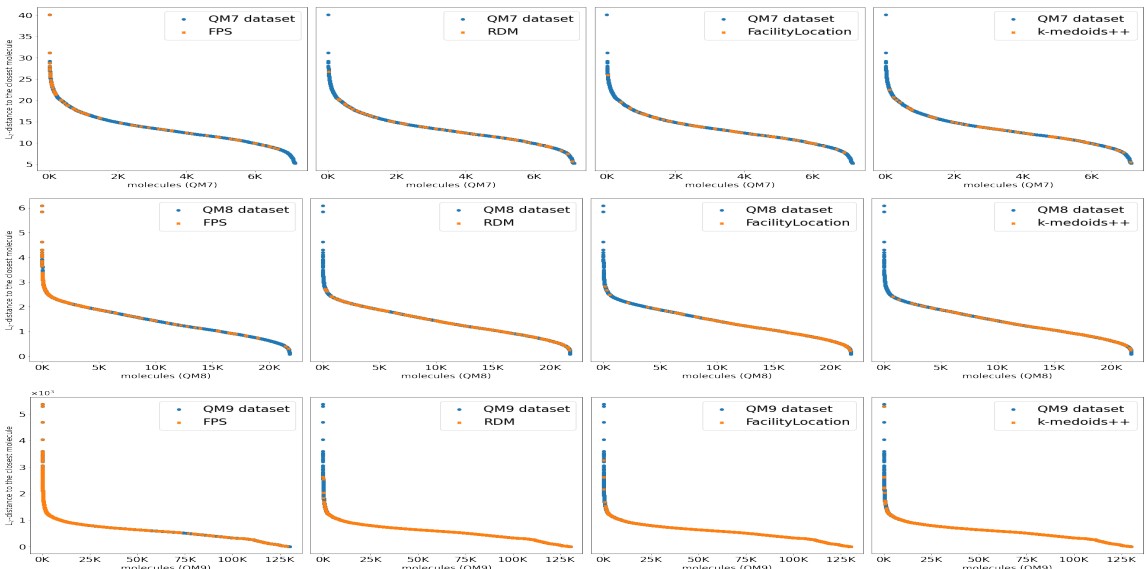

Figure 7: In blue, the Euclidean distances to the nearest neighbour for molecules in QM7 (top row), QM8 (middle row) and QM9 (bottom row). In orange are highlighted the molecules selected with FPS and the other baselines. For each dataset we selected 1% of available data points.

sets selected with the FPS and on sets selected initially with the FPS, the first 2%, and then selected randomly. The figure clearly illustrates that after the FPS has been employed to sample the first 2% of the dataset, the MAXAE of the predictions does not tend to decrease or increase dramatically for larger training set sizes, even if the later samples are selected randomly, independently of the datasets and regression model considered. This fact further suggests that the FPS is mainly beneficial in the low data limit and is strongly connected with the ability of this sampling strategy to select samples at the tail of the data distribution.

### 6.5.3 Importance of the data assumptions

We now highlight the importance of the data assumptions in ensuring that a fill distance minimization strategy leads to a significant reduction of the MAXAE, in correspondence to the theoretical result proposed in Theorem 4. The focus is on Assumption 2, Formula (5), indicating that if two data points have close representations in the feature space, then the conditional expectations of the associated labels are also close. Simply put, this assumption states that if two data points have similar features, their labels are more likely to be similar as well. Therefore, we expect the pairwise distances in the feature and label space to be directly correlated for the experiments to ensure consistency with the theory.

One approach to test this hypothesis on a given dataset is to calculate the Euclidean distance in the feature and label spaces for each pair of points and then calculate Pearson's ($\rho_p$) or Spearman's ($\rho_s$) correlation coefficient (Boslaugh, 2008) to assess the strength and direction of the correlation between the pairwise distances. These coefficients measure how closely correlated two quantities are, with values ranging from -1 to 1. Pearson's coefficient,

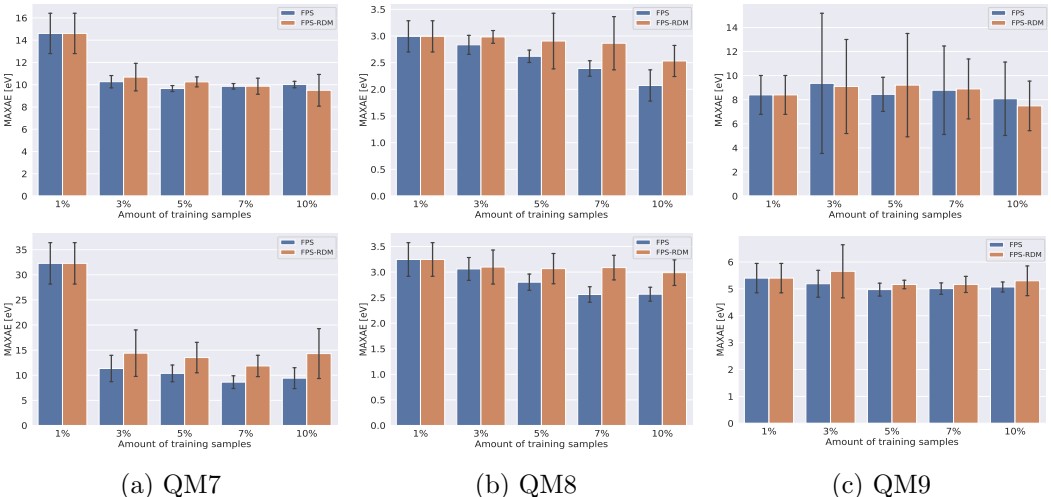

(a) QM7       (b) QM8       (c) QM9

Figure 8: Results for regression tasks on QM7, QM8 and QM9 using KRR (top row) and FNN (bottom row) trained on sets of various sizes and selected with the FPS and with the FPS combined with random sampling after 2% of the available data have been selected. The graphs illustrate the MAXAE of the predictions for each training set size and sampling approach.

captures linear relationships between variables, whereas Spearman's coefficient, measures monotonic relationships regardless of linearity. A positive value indicates a positive correlation, while a negative value indicates a negative correlation. Following along Schober et al. (2018), we define the correlation between the two analyzed quantities to be negligible if the considered correlation coefficient $\rho$ is such that $|\rho| \leq 0.1$, otherwise we consider the correlation positive or negative, depending on the sign of $\rho$.

We test our hypothesis on the data assumption for the experiments analyzed in Section 6.5.1 and illustrated in Figs. 2 and 3. For completeness, we consider both Pearson's ($\rho_p$) and Spearman's ($\rho_s$) coefficient. The computed coefficients are 0.149, 0.216, and 0.272 for $\rho_p$, and 0.281, 0.189, and 0.216 for $\rho_s$, for QM7, QM8, and QM9, respectively. These numbers indicate that in all experiments where the fill distance minimization approach is successful in significantly reducing the maximum prediction error, there is a positive correlation between the pairwise distances of the data features and labels.

Moreover, we also want to show that if the correlation between the pairwise distances in the feature and label space is negligible, the fill distance minimization approach may not lead to a significant reduction in the maximum prediction error. To illustrate this, we perform additional experiments on the QM8 dataset. In these experiments, we examine various labels not yet considered in this work while considering the same data features we previously used. The labels we now consider are the second singlet transition energy (E2), measured in eV, and the first and second oscillator strengths (f1 and f2), measured in atomic units (a.u.). Our computations reveal a Pearson's and Spearman's correlation coefficient of 0.278 and 0.236 for E2, respectively. As for correlations with f1 and f2, Pearson's coefficients are 0.065 and -0.034, respectively, while Spearman's coefficients are 0.098 and -0.036, accordingly. These results suggest a positive correlation between pairwise

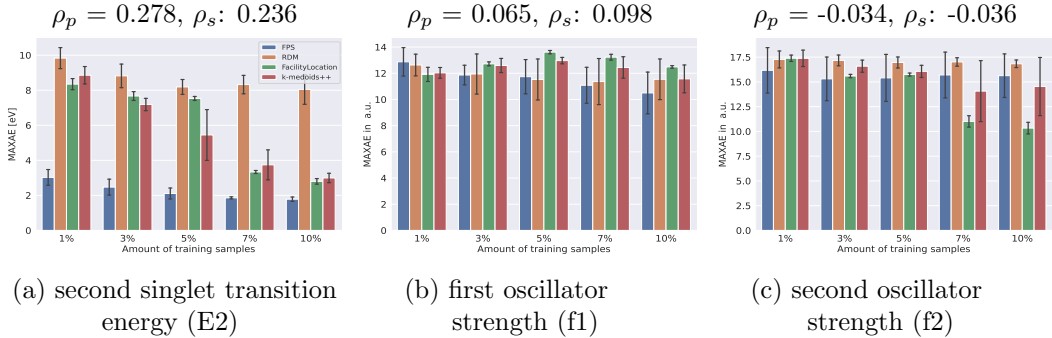

(a) second singlet transition
energy (E2)

(b) first oscillator
strength (f1)

(c) second oscillator
strength (f2)

Figure 9: Results for regression tasks on QM8 considering three different labels: (a) second singlet transition energy, (b) first oscillator strength, (c) second oscillator strength. Regression performed using KRR trained on sets of various sizes and selected with different sampling strategies. The graphs illustrate the MAXAE of the predictions and the legend in (a) applies to (b) and (c) as well. $\rho_p$ and $\rho_s$ are Pearson's and Spearman's correlation coefficients of the data points pairwise distance in the feature and label space.

distances in the feature and label space when considering E2 as the label value, but negligible correlations for f1 and f2. This rejection of our hypothesis for f1 and f2 indicates that our initial assumptions about the data properties may not hold true when considering these two labels. Fig. 9 shows the results for the regression tasks on the QM8 dataset considering E2, f1 and f2 as labels, and using the KRR as regression model. Specifically, Fig. 9b and 9c illustrate the MAXAE of the predictions for the regression tasks with f1 and f2, respectively, and suggest that selecting the training set by fill distance minimization using FPS, does not lead to a significant reduction in the maximum prediction error when the correlation between the pairwise distances in the feature and label space is negligible. On the contrary, Fig. 9a, illustrating the results on the E2 regression task, provides further evidence that the fill distance minimization approach is effective when the correlation is positive, in correspondence to our hypothesis on the data assumption. It is important to note that for the case of the QM8 dataset with f1 or f2 as labels, where the correlation between pairwise distances in the features and label space is negligible, selecting training sets by fill distance minimization approach with the FPS is either comparable or better than randomly choosing the points in terms of the MAXAE of the predictions. Moreover, it is also important to mention that, no benchmark approach can consistently perform better than FPS. For instance, the facility location approach performs best on the f2 regression task for training set sizes of 7% and 10%, but is the worst performing on the f1 regression task for all training set sizes other than 1%.

### 6.5.4 FORCE-FIELD PREDICTION ON THE RMD17 DATASET

In this section, we empirically investigate the effects of minimizing the training set fill distance for multivariate regression tasks on the rMD17 dataset. The label value to predict is the molecular force-field along a molecule's trajectory. We note that for the experiments on the rMD17 we analyze a different range for the size of the training sets, from 0.1% to 1% of the available points, instead of the range 1%- 10%. This change was made because the data points associated with each molecule in the rMD17 are taken from time series and may have

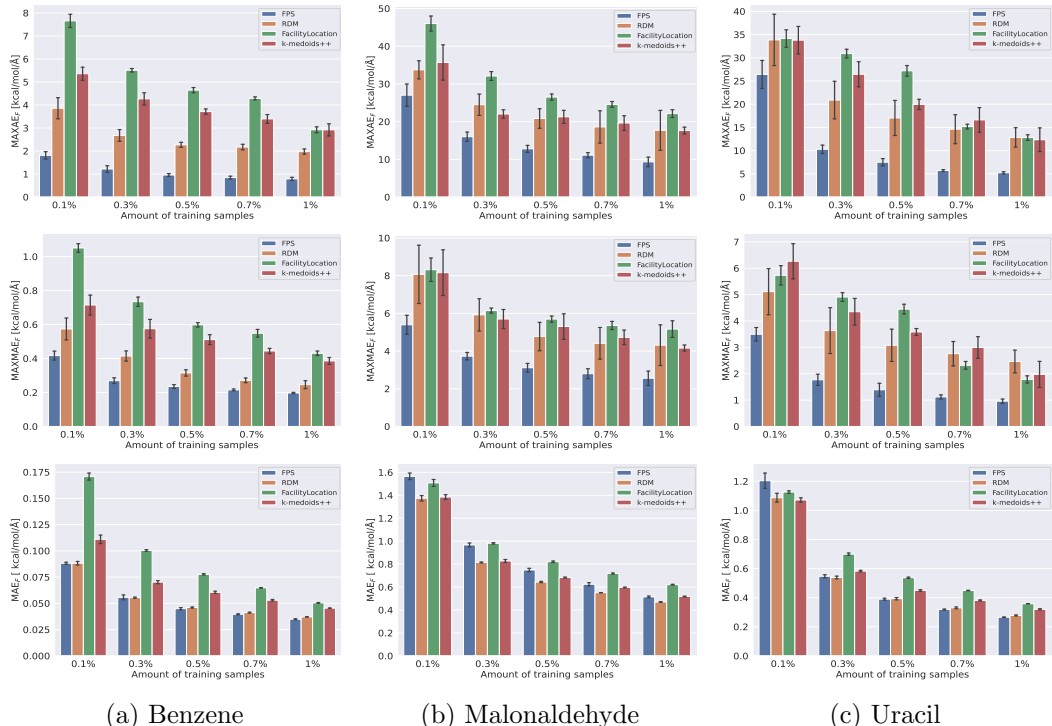

(a) Benzene     (b) Malonaldehyde     (c) Uracil

Figure 10: Results for multivariate regression tasks on trajectories of Benzene, Uracil and Malonaldehyde using GDML trained on sets of various sizes, expressed as a percentage of the available data points for each trajectory, and selected with different sampling strategies. $\text{MAXAE}_F$ (top row), $\text{MAXMAE}_F$ (middle row) and $\text{MAE}_F$ (bottom row) are shown for each training set size and sampling approach.

a high degree of correlation. For this reason the authors of the rMD17 suggest "DO NOT train a model on more than 1000 samples from this dataset" (Christensen and von Lilienfeld, 2020a). Since each trajectory in our analysis consists of 100000 points, limiting ourselves to at most 1% of the available data ensures that we respect the constraint set by the authors. Fig. 10 shows the results for the force-field regression tasks on the trajectories of Benzene, Malonaldehyde and Uracil using GDML. The graphs on the top and middle rows of Fig. 10 illustrate the $\text{MAXAE}_F$ and $\text{MAXMAE}_F$ of the predictions on the unlabelled points. The results suggest that, independently of the trajectory considered, selecting the training set by fill distance minimization using FPS we can perform better than the baselines in terms of these two metrics quantifying the robustness of the model predictions. The graphs on the bottom row of Fig. 10 show the $\text{MAE}_F$ of the predictions. These graphs indicate that selecting training sets with FPS does not drastically reduce the $\text{MAE}_F$ of the predictions on the unlabelled points with respect to the baselines, independently of the trajectory considered, similarly to the experiments with scalar labels. On the contrary, we observe examples where FPS performs worse than one of the baselines, e.g., on the trajectory of Malonaldehyde the FPS performs consistently worse than random sampling. Nonetheless, these experiments suggest that selecting training set by fill distance minimization with the FPS can be beneficial for multivariate regression tasks, increasing models robustness by

reducing of the entry-wise maximum error of the predictions. We remark that the theoretical analysis we propose in Section 4 is limited to regression tasks with scalar label values, and Lipschitz continuous models. Therefore, while the results illustrated in Fig. 10 show promising potential for the FPS in increasing model robustness in multivariate regression tasks, they are not supported by a solid theoretical result.

## 7 Conclusion and Future work

We study the effects of minimizing the training set fill distance for Lipschitz continuous regression models. Our numerical results have shown that, under the given data assumption, using FPS to select training sets by fill distance minimization increases the robustness of the models by significantly reducing the prediction maximum error, in correspondence to our theoretical motivation. Furthermore, we have shown theoretically and empirically that, if we consider kernel regression models, selecting training sets with the FPS also leads to increased model stability. Additionally, we have seen that FPS is particularly advantageous with low training data budgets and argued that there may be more convenient sampling strategies than FPS to select larger amounts of points and improve the average quality of the predictions of a regression model as well.

Based on these remarks, two questions naturally arise: Firstly, how to determine a priori the minimal amount of points to be selected with the FPS? Secondly, how can we modify the FPS to select training sets that can also reduce the average prediction error of a regression model on the data distribution? One possible solution to address the second question is to modify FPS by considering weighted distances. We propose to thereby take the distribution of the data during the sampling process into account, giving higher importance to points in regions of the feature space with higher data density.

## Broader Impact Statement

Minimizing the training set fill distance can be highly beneficial in applications where traditional labelling methods, such as numerical simulations or laboratory experiments, are computationally intensive, time-consuming, or costly, such as in the field of molecular property computations. In such applications, ML regression models are used to predict the unknown labels of data points quickly. However, their accuracy is highly dependent on the quality of the training data. Therefore, careful selection of the training set is crucial to ensure accurate and robust predictions for new points. Our research on minimizing the training set fill distance can be used to identify training sets that have the potential to improve prediction robustness across a wide range of regression models and tasks. This approach prevents the wastage of expensive labelling resources on subsets that may only benefit a particular learning model or task.

## Acknowledgments and Disclosure of Funding

This work was partly supported by the German Federal Ministry of Education and Research (BMBF) within the projects MaGriDo (Mathematics for Machine Learning Methods for

Graph-Based Data with Integrated Domain Knowledge), FKZ 05M20PDB, and DKZ.2R (Rhine-Ruhr Center for Scientific Data Literacy), FKZ 16DKZ2030C.

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
