# OpenReview forum: "On minimizing the training set fill distance in machine learning regression"
_DMLR — Accepted by DMLR_

### Review · Reviewer_QiPK · 2024-04-03

**Recommendation:** 3
**Confidence:** 2

**Summary Of Contributions:**

The authors present a theoretical and empirical analysis of farthest point sampling (FPS), a method for selecting subsets of training data that need to be labeled to maximize the quality of the downstream machine learning model. In this paper, the authors focus their attention on regression models, and a specific measure of model quality -- maximum (expected) prediction error. In their theoretical analysis, the authors prove that the maximum expected prediction error is bounded by factors that depend on the *fill distance* between the set of labeled training data examples and the entire training dataset. Fill distance is defined as the maximal distance between an arbitrary training data example and a labeled data example that is closest to w.r.t. some distance metric (the authors use the L2-norm). In their experimental analysis, the authors evaluate the data selection algorithm on three chemistry-related datasets and two models (kernel ridge regression with the Gaussian kernel and a 3-layer multilevel perception). The results show that the FPS method outperforms the random baseline and two other greedy baselines when maximum prediction error is the model quality metric.

**Strengths:**

**(S1)** The problem of selecting the optimal subset of data to label is important and relevant to the DMLR community.

**(S2)** The proposed theoretical analysis leads to a better understanding of the problem and the behavior of the FPS method w.r.t. the maximum prediction error model quality metric.

**(S3)** The algorithm that the authors study is very simple and easy to implement, making it very applicable.

**Audience:**

Yes

**Broader Impact Concerns:**

The authors provide a broader impact statement and I see no concerns with it.

**Claims And Evidence:**

The provided claims are fair, fairly nuanced, and theoretically well-supported. The empirical support could have been stronger with a wider array of diverse datasets and models.

**Datasets And Benchmarks:**

N/A

**Extended Submissions:**

A quick Google search revealed that the submission is an extension of the paper entitled "Investigating minimizing the training set fill distance in machine learning regression" which was published at the 2023 DMLR workshop. I haven't seen this explicitly stated (although perhaps this was an issue with the Open Review submission form not explicitly asking about this?)

Based on my light-weight comparison, the original paper is extended with the following parts:
* Analysis of the FPS method for the kernel ridge regression model
* Addition of one more dataset to the experimental evaluation
* Empirical analysis of the importance of the assumptions introduced earlier in the paper

Overall, this appears like it does extend the original work with ~30% of raw content. That said, the original paper was published in a workshop, and I feel that it would need a significant qualitative improvement (not just quantitative) to be published in a journal.

**Limitations:**

**(L1)** Even though the introduction does a fair job of positioning the paper in the context of prior work, the specific angle the authors chose (i.e. using the maximum prediction error as the target quality metric) could be better motivated. On one hand, the authors' contribution fundamentally relies on this metric and it is the only metric where the FPS method dominates the baselines. On the other hand, its motivation is mentioned only in one sentence of the introduction and described only as a "helpful metric in various application fields". This does not leave the impression of an important problem that the community really needs to address.

**(L2)** The theoretical analysis could benefit from leveraging concrete examples and/or visualizations. This would improve the readability of the paper by helping the reader build intuition without necessarily having to parse out all the proofs.

**(L3)** Several sections are hard to follow (particularly sections 5 and 6.5.2) due to a lack of introductory text connecting them to the wider story of the paper. A rewrite of these sections would help the readability by explaining their purpose for the paper and the information the reader is supposed to take away from them.

**(L4)** The empirical evaluation (also one of the stated contributions of the authors' work) could be improved by introducing more diverse datasets and models. The two cited papers that perform an empirical analysis of FPS for regression evaluate this method on 9-10 datasets while the authors evaluate only on 3. In terms of models, I understand that the authors wanted to focus on Lipshitz-continuous models, but perhaps the FPS could go beyond this and be applicable to other regression models. A wider array of datasets and models would provide more compelling evidence for the usability of FPS in a broader variety of scenarios.

**Requested Changes:**

**(C1)** (Section 1) I suggest providing a much stronger motivation explaining the importance of the maximum prediction error metric. Who uses it? What are the scenarios and why are they important? What happens in case the maximum prediction error is large? Instead of a single sentence, this should be extensively explained, perhaps even in a whole paragraph.

**(C2)** (Sections 3, 4, and probably 5) I would augment the theoretical sections by adding some running example(s) with real numbers and/or visualizations of the key concepts. A non-exhaustive list of candidates for this type of additional explanation: maximum expected absolute error (eq 1), fill distance, assumptions, the bound...

**(C3)** (Section 5) The purpose of this section is unclear. I would add a solid introductory paragraph connecting it to the broader context and explaining the purpose of this analysis.

**(C4)** (Section 5) This claim is a bit stretched -- "Thus, if we restrict the spectrum of the selection strategies of interest to those that run in polynomial time, as it is the case when we work with large datasets, the FPS provides ***optimal*** solutions to both the problems in (11) and (20)." I assume that what you want to say is something along the lines of "that this is the best possible PTIME approximation of the optimization problem". There's no need to claim that the solution produced by FPS is optimal. Even with the additional caveat it feels like a stretch.

**(C5)** (Section 6) I think the experimental section should be substantially augmented by adding a greater variety of datasets and models. The empirical validation is one of the stated contributions of the work, which makes it very important that this part is very well done with plenty of empirical evidence of the wide applicability of this method that would give confidence to the reader that they should apply it to their own problem as well.

**(C6)** (Section 6.5.2) Same as **(C3)**, the purpose of this section is unclear. What are the hypotheses being evaluated? What kind of result are we expecting in order to validate these hypotheses? Section 6.5.3 does a better job at this. A similar approach could be applied to section 6.5.2.

**(C7)** (various minor comments)
1. In definition 1, there is a sentence "Put differently, we have that ***any*** point $x \in \mathcal{D}\_{\mathcal{X}}$ has a point $x\_j \in \mathcal{L}\_{\mathcal{X}}$ not farther away than $h\_{\mathcal{L}\_{\mathcal{X}}, \mathcal{D}\_{\mathcal{X}}}$". Based on my understanding of equation 2, perhaps you wanted to say "each" instead of "any"?
2. In section 6.5.1, there is a sentence "The bottom row of Fig. 3 illustrates the condition numbers of the non-regularized kernels and shows that, if no regularization is applied, for the QM7 and the QM8 there is an ***exponential difference*** between the condition numbers of the matrices obtained with the FPS and those obtained using the benchmark strategies, as expected from (19)." Firstly, slightly too long of a sentence. But, more importantly, what is an exponential difference? Perhaps you wanted to say "an order of magnitude difference"?

**(C8)** (optional) Would it be possible to add an analysis and describe a methodology that could help answer the question "Can FPS work well on my dataset?" Can the specific Lipschitz constants in equation 7 be translated to some measurable parameters of the dataset and thus enable the potential user of this method to know ahead of time what benefit could they achieve? I mark this request as optional because I think it would be very nice to have but I'm unsure how difficult it is, and if it could simply be added to this paper or if it would require a separate project to answer.

---

### Review · Reviewer_xHeE · 2024-06-23

**Recommendation:** 3
**Confidence:** 3

**Summary Of Contributions:**

This paper theoretically and empirically studies the effect of minimizing the fill distance through farthest distance selection (FPS) for the regression problem. The author first defines the problem of robust prediction by minimizing the maximum of the expected prediction error for the label obtained in the trained regression model. Then, they introduce the fill distance of the training subset and prove an upper bound for the maximum expected prediction error of the Lipschitz continuous regression model, which is a linear function of the fill distance of the training set plus the label uncertainty and the maximum error achieved on the training set.

To minimize the maximum expected prediction error, they choose to minimize this upper bound, which is, in essence, minimizing the fill distance. This minimization problem is NP-hard, but they show that it can be solved via FPS within polynomial time whose maximum prediction error is at most 2 times the minimization.

They then use Kernel Ridge Regression as an example to show the effectiveness of FPS. In KRR, the numerical stability can be represented via thee condition number of the kernel matrix. The maximal eigenvalue of kernel matrix can be upper bounded trivially, while the minimal one can be lower bounded by a function of the separation of the dataset. Therefore, to increase the robustness, one needs to maximize the separation of the dataset, which can also be approximated done by FPS with a factor of at most 2.

They finally did experiments on QM7/8/9 dataset using Kernel Ridge Regression(KNN) and Feedforward Neural Networks(FNN). They compared FPS with random sample, the facility location algorithm, and the K-medrod++. They showed that FPS can effectively and significantly reduce the maximum absolute error while maintaining the same level of mean absolute error, showing that FPS can effectively select a training set which facilitates the robustness of a trained regression model.

**Strengths:**

1. The Theorem 4, as the main theoretical result of this paper, is solid, and the proof is correct from my perspective. The assumption 2 and 3 are proper.

2. The experiment's result is solid because the maximum absolute error for FPS is significantly smaller than that of the baseline methods.

**Audience:**

Yes

**Broader Impact Concerns:**

/

**Claims And Evidence:**

The claims made in the submission are supported by either theoretical proof or experiments.

**Datasets And Benchmarks:**

The authors describe the dataset they use thoroughly and in very detail. The dataset is public, and they also offer the codebase they used.

**Extended Submissions:**

/

**Limitations:**

1. The KRR section is more like a review or an explanation of why FPS can improve numerical stability. It mainly combines the results of a prior paper to show that the minimal eigenvalue of the kernel matrix can be lower bounded by the separation and that FPS can approximately solve the separation minimization problem. I think this section does not add much novelty to this paper since the authors do not prove those results. All results in sections 4 and 5, except Theorem 4, are already known, which makes this paper less novel and more like a new explanation of the old facts (and algorithms).

2. Theorem 4 upper bounds the minimum of the maximum of expected prediction reward by a multiple of fill distance plus two additive terms (the label uncertainty and the max error of training set). In the following section, the authors consider minimizing the fill distance without discussing the influence of those two additive error terms. Although they claim that they are negligible, they did not prove it or reveal it in the experiments.

**Requested Changes:**

/

---

### Review · Reviewer_pkzG · 2024-06-24

**Recommendation:** 3
**Confidence:** 3

**Summary Of Contributions:**

This paper aims to find a new way to better select training data set so that the maximum expected error associated to a trained regression model is minimized. The authors considered Lipschitz learning algorithms such as Gaussian Kernel regression and Feed-Forward neural network and applied fastest point sampling method to choose the training set. Due to their Lipschitz property, the authors were able to prove a theoretical upper bound of the  maximum expected error.

    Based on the experiments over 3 data set, the authors showed the advantages of their proposed methods compared to baselines; moreover, they analyzed the properties of the data set which made their proposed method perform better, and this property is consistent with their major assumptions made in their main theorem.

    It is a new attempt to consider minimizing error bound from a theoretical point of view for regression model with detailed analysis and experiments.

**Strengths:**

1. The topic is new and it seems that this is one of the first a few papers to consider selecting a training set to minimize the maximum expected error for regression ML models;

2. There are both mathematical proof of the upper bound of the maximum expected error; the authors also analyzed the stability of Gaussian kernel regression using their proposed framework. The experiments supports the theoretical results and the authors also analyzed under what conditions their proposed framework works better than others.

**Audience:**

Yes

**Broader Impact Concerns:**

There seems no obvious such concern in this paper.

**Claims And Evidence:**

Yes. At least based on the experiments made in the paper, they are supported.

**Datasets And Benchmarks:**

Yes; the authors did a detailed analysis of their proposed experiments, and provided GIT repo for reproducing their results in Section 6.

**Extended Submissions:**

It seems that a very similar version was accepted as poster in DMLR Workshop 2023; I did not find the link, so I searched the public paper having the same title as the poster, and it seems that the 2 papers are almost essentially the same except the order of the contents.

**Limitations:**

1. The model considered in the paper is useful but mainly constrained for Lipschitz continuous ML algorithms; this helps to make the proof of
Theorem 4 relatively easy to get, but also make the theorem not general enough, make the proof a little bit too straightforward. Another constrain is that only KRR and FNN are analyzed while other very useful regressions models are not in the scope.

2. The authors proposed to considered to minimize maximum expected error instead of the average error as in the classification case in previous work. However, average error can be useful for many cases, the authors should consider it if possible, or discuss if there is any possible technical difficulty to obtain an upper bound of the average error or mean square error.

3. In the comments after Theorem 4 and it proof, the authors discussed the importance of the labels uncertainty $\epsilon$  and the maximum error on the labelled data $\epsilon_{L}$; it seems that the authors did not discuss them in their experiments. But as shown in Theorem 4, these quantities are important for estimating the upper bound of the maximum expected error.

**Requested Changes:**

Critical:
If possible, please add more details and mathematical analysis of the stability analysis of FNN models with FPS.

If possible, please add some discussions on labels uncertainty $\epsilon$  and the maximum error on the labelled data $\epsilon_{L}$ in the experiments.

simply strengthen the work:

In the paper, the authors me mentioned that the data set need to satisfy  Assumption2 which means that similar features imply similar regression results. Maybe the authors could also consider some feature transformation and embedding before this step so that this framework is more general.

Another point is that maybe the authors can first apply some clustering methods to group the data points, and them within each group apply their proposed framework; which may help to increase their accuracy for MAE.

---

### Review · Reviewer_Duez · 2024-06-26

**Recommendation:** 3
**Confidence:** 2

**Summary Of Contributions:**

This work investigates the use of Farthest Point Sampling (FPS) for selecting small training sets from large pools of unlabelled data in regression tasks. The authors showed that by aiming to minimize the fill distance of the selected set, FPS can significantly reduce the maximum prediction error of various regression models. The authors derive an upper bound for the maximum expected prediction error that linearly depends on the fill distance and empirically validate their approach using two regression models across three datasets. The results demonstrate that FPS not only outperforms alternative sampling methods but also enhances model stability, particularly in Gaussian kernel regression.

**Strengths:**

- (S1) This paper provides an upper bound on the expected error in regression problems when applying farthest point sampling in Gaussian kernel ridge regression. The theoretical proof of this upper bound is valid, clear, and demonstrates numerical stability.

- (S2) The paper successfully demonstrates that farthest point sampling can significantly reduce computation time while maintaining good predictive accuracy and model fitting.

- (S3) This paper provides an upper bound on the expected error in regression problems when applying farthest point sampling in Gaussian kernel ridge regression. The theoretical proof is valid, clear, and demonstrates that the numerical stability for this specific kernel structure can be bounded.

- (S4) Additionally, the paper successfully demonstrates that farthest point sampling can significantly reduce computation time while maintaining good predictive accuracy and model fitting.

**Audience:**

Yes

**Broader Impact Concerns:**

The broader impact concerns are presented in the paper and addressed sufficiently.

**Claims And Evidence:**

Yes, the claims made in the submission are supported by accurate, convincing, and clear evidence.

**Datasets And Benchmarks:**

Yes, the detail of data collection and organization was clearly stated in the paper.

**Extended Submissions:**

One of my concerns is that a significant portion of the content, including algorithms, datasets, and comparative methods, is similar to the "Investigating Minimizing the Training Set Fill Distance in Machine Learning Regression" paper published in DMLR 2023. According to DMLR's acceptance criteria, a broader scope of data analysis and simulation results is needed to meet the "contain a minimum of 30% additional content" requirement.

Specifically, here is a list of necessary additions to fulfill the extended requirements:

1. Additional datasets beyond QM7, QM8, and QM9.
2. Numerical experiments illustrating the bounding effect of the expected error.
3. Sensitivity analysis to explore how variations in hyperparameters affect the performance and stability of the proposed method.

Including these additional analyses and discussions will significantly enhance the submission and help meet the criteria for the extended content requirement, ultimately providing a more comprehensive and impactful contribution to the field.

**Limitations:**

- (L1) In this paper, the author proves that the expected error can be upper bounded. Therefore, I expected a simulation study demonstrating the bounded effect of the expected error. However, I couldn't find any evidence in the data analysis that the expected error is upper bounded.

- (L2) Another concern is that the authors only show the theoretical properties, such as numerical stability and the predictive error upper bound, for kernel ridge regression with a specific kernel structure, namely the Gaussian kernel. However, in most kernel regression problems, the choice of kernel is sensitive to the choice of the kernel function.

- (L3) It would be beneficial to compare the performance of FPS when applied to regression methods using other common kernel choices, even if numerical stability may not be maintained. This comparison would provide insights into the versatility of FPS beyond its application in Gaussian kernel ridge regression.

**Requested Changes:**

Major Changes:

- Following (L1), an experiment separate from the data analysis, demonstrating the bounding effect of the expected absolute error, would be very helpful. In the data analysis, since we don't know the ground truth, this additional experiment is necessary.

- Similarly, an visualization or toy example illustrating the bounding effect $\lambda_{\min}(\mathbf{K}_{\mathcal{L}}) \geq ...$, will be helpful as well.

- Following (L2), could the authors address the potential generalizability of the theoretical properties of farthest point sampling (FPS)? For example, if the Matérn kernel or Gamma-exponential kernel is used, would the numerical stability still be bounded?

Minor Changes:

- The inequality (5), the $\lambda_p$ should be defined before/after the assumption statement.